# Recurrent circuitry is required to stabilize piriform cortex odor representations across brain states

Kevin A Bolding, Shivathmihai Nagappan, Bao-Xia Han, Fan Wang, Kevin M Franks*

Department of Neurobiology, Duke University Medical School, Durham, United States

**Abstract** Pattern completion, or the ability to retrieve stable neural activity patterns from noisy or partial cues, is a fundamental feature of memory. Theoretical studies indicate that recurrently connected auto-associative or discrete attractor networks can perform this process. Although pattern completion and attractor dynamics have been observed in various recurrent neural circuits, the role recurrent circuitry plays in implementing these processes remains unclear. In recordings from head-fixed mice, we found that odor responses in olfactory bulb degrade under ketamine/xylazine anesthesia while responses immediately downstream, in piriform cortex, remain robust. Recurrent connections are required to stabilize cortical odor representations across states. Moreover, piriform odor representations exhibit attractor dynamics, both within and across trials, and these are also abolished when recurrent circuitry is eliminated. Here, we present converging evidence that recurrently-connected piriform populations stabilize sensory representations in response to degraded inputs, consistent with an auto-associative function for piriform cortex supported by recurrent circuitry.

*For correspondence:
franks@neuro.duke.edu

Competing interests: The authors declare that no competing interests exist.

## Introduction

Recognition occurs at the interface of perception and memory: it requires being able to reliably identify a familiar object, even when the stimulus is noisy or degraded, or when behavioral states change. Sensory systems must, therefore, be able to generate representations of the stimuli that are robust to changes in input or ongoing brain activity. Theoretical studies have shown that this retrieval can occur in recurrent neural networks through processes called auto-associative recall, attractor dynamics, or pattern completion (*Hopfield, 1982*; *Marr, 1971*; *Haberly and Bower, 1989*; *Hasselmo et al., 1990*; *Treves and Rolls, 1994*; *Eichenbaum, 2004*; *Rolls, 2013*; *Guzman et al., 2016*; *Chaudhuri and Fiete, 2016*). Phenomenological evidence for pattern completion has been described in various recurrent networks, including hippocampus (*Nakazawa et al., 2002*; *Wills et al., 2005*; *Neunuebel and Knierim, 2014*), piriform cortex (*Chapuis and Wilson, 2012*), and neocortex (*Suzuki and Gottlieb, 2013*; *Carrillo-Reid et al., 2016*; *Inagaki et al., 2019*). However, despite recurrent collateral connections being a defining feature of auto-associative networks, whether these phenomena indeed require recurrent connectivity has not been demonstrated.

Odors are initially encoded as combinations of co-active glomeruli in the olfactory bulb (OB). This elemental odor code is then integrated in the piriform cortex (PCx) where a synthetic or configural representation of the odor object is first formed (*Gottfried, 2010*; *Wilson and Sullivan, 2011*; *Howard and Gottfried, 2014*) but see *Barwich, 2019*. Projections from OB to PCx are diffuse and overlapping (*Sosulski et al., 2011*; *Ghosh et al., 2011*; *Miyamichi et al., 2011*) allowing individual principal neurons in PCx to integrate inputs from different and possibly random subsets of glomeruli. The diffuse projections from OB to PCx activate odor-specific ensembles of neurons distributed

across PCx [*Stettler and Axel, 2009*; *Roland et al., 2017*] whose concerted activity encodes odor identity (*Miura et al., 2012*; *Bolding and Franks, 2017*). Alone, these afferent inputs would render PCx odor representations wholly dependent on their inputs from the OB. However, principal cells make excitatory synaptic connections onto other PCx cells with sparse but uniform connection probabilities across millimeters of PCx, forming an extensive recurrent network, similar in synaptic organization to hippocampal CA3 (*Johnson et al., 2000*; *Franks et al., 2011*; *Hagiwara et al., 2012*; *Guzman et al., 2016*). Selective strengthening of recurrent synapses between co-responsive PCx neurons can lead to the formation of cortical odor cell assemblies that can generate stable odor representations when OB inputs are noisy or degraded (*Hasselmo et al., 1990*; *Ambros-Ingerson et al., 1990*; *Haberly, 2001*).

PCx populations can demonstrate pattern separation or pattern completion-like responses depending on recent training history (*Chapuis and Wilson, 2012*). However, these and other major observations regarding associative functions and odor-evoked activation patterns in PCx were obtained in either ex vivo (*Hasselmo and Barkai, 1995*; *Barkai et al., 1994*) or anesthetized preparations (*Barnes et al., 2008*; *Chapuis and Wilson, 2012*). Odor responses in both OB and PCx are reported to depend strongly on global brain state (*Murakami et al., 2005*; *Kato et al., 2012*; *Rinberg et al., 2006*). Previous studies, using different methods, have reported that OB responses increase (*Kato et al., 2012*; *Rinberg et al., 2006*) or remain constant (*Kollo et al., 2014*) under anesthesia. Curiously, we found that OB responses reliably degraded shortly after a single injection of a ketamine-xylazine anesthetic cocktail. We leveraged this observation to ask how PCx odor ensembles change when odor input is degraded. Interestingly, we found little difference in the quality of PCx representations across brain states, suggesting that PCx circuits may transform noisy or degraded OB inputs into stable cortical odor representations. Using a novel transgenic line and selective disruption of PCx output, we find that intracortical synaptic connections within PCx are required for this stabilization, consistent with the crucial role for recurrent connectivity in implementing cortical pattern completion.

## Results

### Odor responsivity is state-dependent in OB but not PCx

We simultaneously recorded spiking activity in populations of OB mitral cells and layer II neurons in PCx using 32-site silicon probes in head-fixed mice before and after inducing ketamine/xylazine anesthesia (k/x; *Figure 1a*). Anesthesia induced pronounced changes in respiration patterns, oscillatory activity, and spontaneous spiking in both OB and PCx (*Figure 1*, b and c; *Figure 1—figure supplement 1*); see also [*Murakami et al., 2005*; *Rinberg et al., 2006*; *Fontanini and Bower, 2005*; *Li et al., 2012*]. Previous reports found reduced OB firing rates under anesthesia, though estimates of spontaneous OB firing rates vary widely (*Rinberg et al., 2006*; *Kollo et al., 2014*; *Shusterman et al., 2011*). Our measures of spontaneous OB activity (n = 187 cells; awake: (mean ± st.dev.) 6.58 ± 6.47 Hz; k/x: 2.62 ± 4.37 Hz) are similar to those observed using whole-cell recordings in vivo by Kollo et al. (n = 60 cells, awake: (mean ± st.dev.) 4.1 ± 6.7 Hz; k/x: n = 84 cells, 2.7 ± 4.6 Hz) (*Kollo et al., 2014*), but are substantially lower than those made with single extracellular electrodes by Rinberg et al. (n = 11 cells; awake: (mean ± st.dev.) 27.4 ± 9.7 Hz; k/x: 9.3 ± 4.4 Hz) (*Rinberg et al., 2006*). These differences likely stem from improved identification of lower firing-rate units using multi-site extracellular electrodes, longer-duration recordings, and improved spike-sorting algorithms.

We examined odor-evoked spiking activity in individual cells in OB and PCx during the first sniff after odor delivery (*Figure 1*, d and e). OB neurons were less responsive under anesthesia, with many fewer significantly activated or suppressed neurons, and increased lifetime and population sparseness (*Figure 1*, f and g). We ensured that this effect was indeed due to changes in spiking activity and not to unit drift or instabilities over the course of the recording (*Figure 1—figure supplement 2*). Yet, despite reduced input from OB, PCx responsivity shifted only subtly and, in fact, toward greater activation (*Figure 1*, h and i; *Figure 1—figure supplement 3*), due largely to an increase in signal-to-noise ratio as spontaneous activity levels in PCx dropped under anesthesia. Individual cell-odor pair responses rarely switched from significantly activated to suppressed or vice versa (OB: act. to supp.: 0.98% of activated responses.; supp. to act.: 3.7% of suppressed responses;

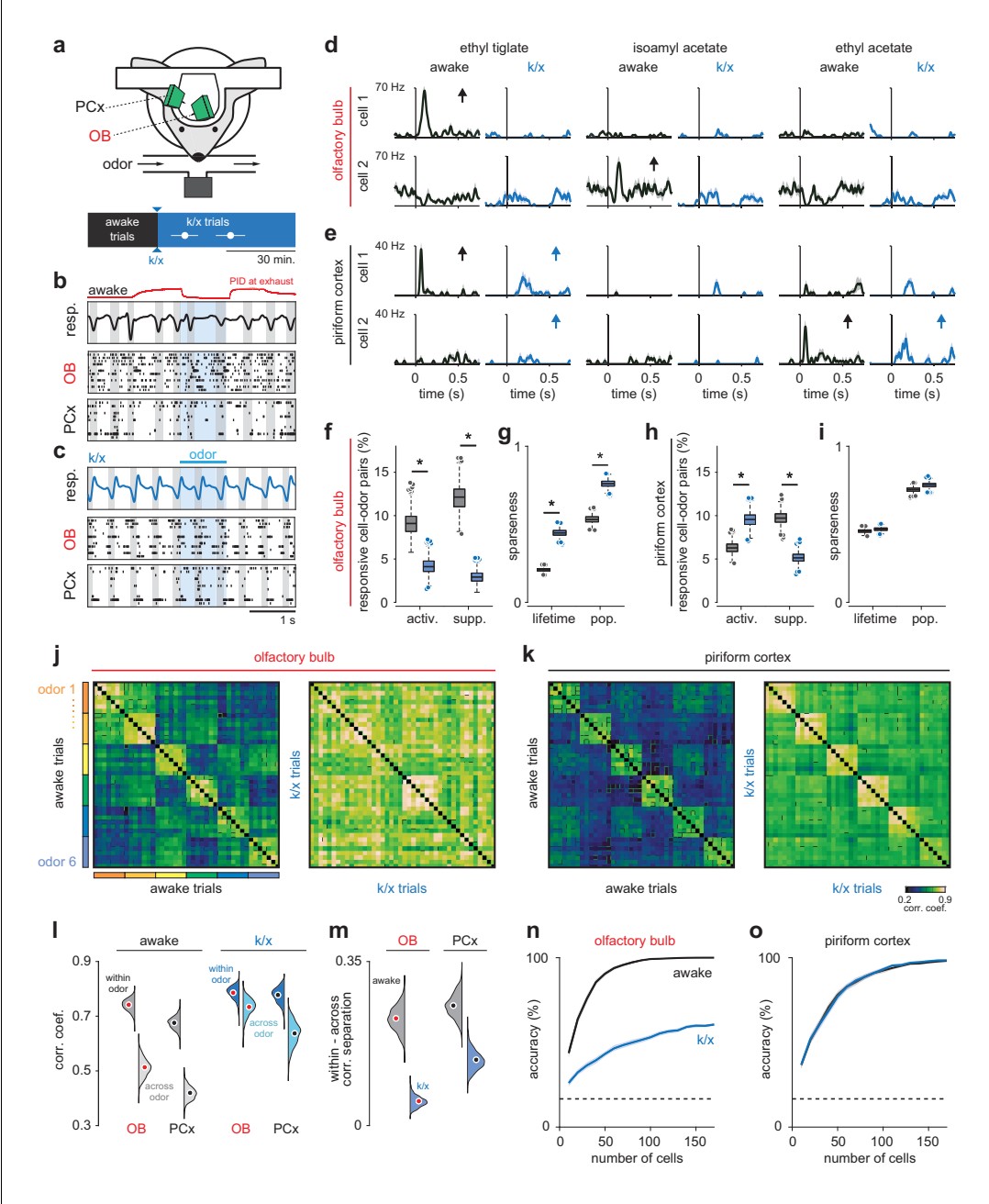

**Figure 1.** Odor responsivity is state-dependent in OB but not PCx. (a) Recording and experimental schematics. Timing of first and last trials designated as anesthetized for analysis, starting from the onset of behavioral indicators of anesthesia and continuing for 10–15 presentations of all odors (n = 12 experiments in six mice). Error bars are mean ± SEM. (b) Example recording showing irregular respiration (*top*) and desynchronized spiking in awake OB (*middle*) and PCx (*bottom*). Ticks indicate action potentials and each row represents a different cell. Gray shading indicates inhalation, blue shading indicates odor delivery, red trace is an example PID trace monitored at the exhaust port of the switched final delivery valve. (c) Same recording as b after k/x injection. Respiration becomes rhythmic and spontaneous spiking in OB and PCx slows and becomes entrained to respiration under anesthesia. (d) Example OB cell-odor pair responses during awake and k/x trials showing loss of significant odor responses under anesthesia. Arrows indicate significant responses. (e) Example PCx cell-odor pair responses during awake and k/x trials showing preserved odor responses under anesthesia. Arrows indicate significant responses. (f) Percent of OB cell-odor pairs with significant activation or suppression (n = 187 cells, 12 experiments). Asterisks indicate p<0.05 in bootstrap difference test (Activation: p=0.007; Suppression: p<0.001). Boxes indicate quartiles and whiskers indicate ± 2.7 st. dev. from mean. Data points outside this range are shown as circles. (g) Lifetime and population sparseness in OB (bootstrap difference test, p<0.05) across all cells and odors. Asterisks indicate p<0.05 on bootstrap difference test. Lifetime: p<0.001; Population: p<0.001. (h–i) As in f-g, but in PCx (n = 640 cells, 11 experiments). Activation: p=0.003; Suppression: p<0.001; Lifetime: p=0.45; Population: p=0.22. (j–k) Trial-by-trial pseudopopulation correlation matrices sorted by odor within OB (j) and PCx (k) in awake and anesthetized conditions. Odors are (1) ethyl butyrate, (2)

*Figure 1 continued on next page*

*Figure 1 continued*

isoamyl acetate, (3) 2-hexanone, (4) hexanal, (5) ethyl tiglate, (6) ethyl acetate. (I) Bootstrapped distributions of within- and across-odor trial-to-trial correlations in OB and PCx computed by sampling cells with replacement and computing the mean correlation within and across odors 1000 times. Means are shown as filled circles. (m) Separation (mean within-odor correlations - mean across-odor correlations) of odor representations in OB and PCx over 1000 bootstrap iterations. (n–o) Odor classification accuracy as a function of pseudopopulation size in OB (n) and PCx (o) in awake and anesthetized states using a multiclass linear support vector machine. Mean ± 95% bootstrapped confidence intervals.

The online version of this article includes the following figure supplement(s) for figure 1:

**Figure supplement 1.** State-dependent changes in respiration, local field potential and spontaneous spiking.
**Figure supplement 2.** Criteria for maintaining sorted unit identity across states.
**Figure supplement 3.** Odor response characteristics during awake and anesthetized trials.
**Figure supplement 4.** Degraded OB decoding due to weaker odor-evoked responses.
**Figure supplement 5.** Trial-trial correlation changes after anesthesia reflect contributions of 'background' patterns and reduced reliability of OB odor responses.
**Figure supplement 6.** Similar outcomes for simultaneously recorded PCx populations and pseudopopulations.
**Figure supplement 7.** Background activity or non-specific odor responses do not significantly impair OB decoding under k/x anesthesia.

PCx: act. to supp.: 0.81%; supp. to act: 5.0%). Tuning curves for neurons in either region did not exhibit an overall tendency toward either maintained ordering or reconfiguration such that there was only a modest, positive average correlation between individual neuron tuning curves for awake and anesthetized responses (OB: 0.13 ± 0.47; PCx: 0.16 ± 0.48). Although we did find some reliable OB responses after k/x injection, consistent with other anesthesia regimes (*Yokoi et al., 1995*; *Dhawale et al., 2010*; *Buonviso et al., 1992*), the reduced OB odor responsivity we observed was unexpected given that previous studies have reported larger and longer-duration odor responses under k/x anesthesia (*Kato et al., 2012*; *Rinberg et al., 2006*). These differences could result from differences in the anesthesia methods, recording methods, odorant concentrations, or behavioral states (please see Discussion). In our experiments, the loss of OB responsivity was largely due to decreased odor-evoked spiking under k/x anesthesia, rather than an increase in response variability or increased baseline firing (*Figure 1—figure supplement 4*). Crucially, however, we found that PCx responsivity was preserved even though OB responsivity was consistently suppressed.

To examine odor responses at the population level we constructed pseudopopulation vectors of OB or PCx firing rates for each odor trial and measured the similarity of responses using trial-to-trial correlations within or across odors (*Figure 1j-l*). In these and other analyses (except those presented in Figure 6), we excluded the first ~5 trials to minimize the contribution of rapid sniffing in response to the first few odor presentations, and we used an equivalent number of trials during stable anesthesia, shortly after initial induction (*Figure 1a*). Responses in both regions became more correlated under anesthesia (*Figure 1j-l*). This was due primarily to a subset of cells in OB and PCx that exhibited weak and stereotyped responses to all odors under k/x anesthesia, driving up both within- and across-trial correlations, although this was especially pronounced in OB (*Figure 1—figure supplement 5*). However, even as all responses became more correlated, the difference between within- versus across-odor response correlations decreased significantly more in OB than PCx (*Figure 1m*; *Figure 1—figure supplement 5i*). Greater inter-odor response separation in PCx than OB under anesthesia was also observed when comparing population vector correlations for simultaneously recorded OB and PCx neuron populations (*Figure 1—figure supplement 6, a and b*).

Next, we determined how accurately odors could be identified from the spiking activity of populations of OB and PCx neurons. To do this, we trained and tested classifiers on either awake or anesthetized odor trials from pseudopopulations of OB or PCx neurons. Decoding accuracy was markedly worse under anesthesia in OB (*Figure 1n*), whereas responses in PCx were decoded equally well in either state (*Figure 1o*). OB response decoding was not statistically significantly improved after accounting for changes in baseline firing rate or the imposition of odor-dependent but non-odor-specific 'background patterns' that may emerge under k/x anesthesia (*Figure 1—figure supplement 7*), indicating that OB decoding degraded due to weakened odor responses rather than the imposition of a global background activity pattern. Finally, classifier performance using individual PCx experiments with simultaneously-recorded populations of up to 60 cells did not differ significantly from accuracies for size-matched pseudopopulations (*Figure 1—figure supplement 6, c and d*).

## PCx stabilizes odor representations across activity regimes

These data indicate that PCx maintains odor responsivity when OB input is degraded, but not whether the cortical odor representations themselves are preserved across states. To address this question, we focused first on cells that had statistically significant increases in odor-evoked spiking. We defined activated cell-odor pairs as either *robust* if they were activated in both regimes or as *state-specific* if they were only activated in awake or only in anesthetized states (*Figure 2a and b*). There were many more *robust* responses in PCx than OB (OB, 21/102 total awake responses; PCx, 118/237 total awake responses, *Figure 2c*, top). To avoid simply classifying responses as 'activated' or 'not-activated' according to an arbitrary statistical threshold, we again considered single-trial population responses using spike rates for all OB and PCx neurons. Compared to OB, PCx population responses to the same odor were more correlated across states (*Figure 2d*) and were more separable from responses to other odors (*Figure 2e*), indicating again that population odor representations are better preserved across states in PCx.

To determine how similar odor-evoked activity patterns were across states, we trained a classifier using responses recorded in the awake state and tested the classifier on responses recorded under anesthesia. Cross-state decoding was better in PCx than OB (*Figure 2f*), indicating that PCx could extract and selectively represent stimulus-specific information from partial and noisy OB input. Nevertheless, cross-state decoding using spike counts was relatively poor in both OB and PCx, and Principal Components Analysis (PCA) indicated that state accounted for most of the variance in responses across states in both regions (*Figure 2—figure supplement 1*). Thus, if pattern stability is a reflection of overlap in a low-dimensional neural activity space, then neither region maintained similar responses across states. We therefore considered an alternative model, in which a downstream decoder could adapt to overall state-dependent changes and still maintain the ability to distinguish stimuli. To explore this possibility, we used demixed PCA to isolate the stimulus-specific features of the population responses (*Kobak et al., 2016a*). This analysis revealed that OB responses to different odors were clearly separable and responses to the same odor overlapped partly in awake and anesthetized states (*Figure 2g*). However, in PCx responses to the same odor overlapped almost completely , indicating that the stimulus-specific features of the cortical odor response are almost identical across states and better preserved than their inputs (*Figure 2h and i*). Thus, regardless of whether downstream processing is fixed or adaptive, PCx actively transforms degraded OB output to represent the stimulus more faithfully than the input it receives.

## Robust PCx representations derive from short-latency OB responses

To generate stable cortical odor representations using degraded and noisy OB input, PCx must over-represent the impact of the few *robust* OB responses. What features of the OB response does PCx use to selectively extract this information? Peak firing rate distributions of *robust* and *state-specific* OB responses were broad and overlapped substantially in either regime (*Figure 3*, a-e), and though robust responses appeared slightly larger, on average this difference was not statistically significant. Instead, *robust* OB responses had significantly shorter latencies than *state-specific* responses (*Figure 3f*). Thus, PCx could over-represent early inputs to produce a stable output (*Bolding and Franks, 2018*). However, *robust* responses in PCx not only had shorter latencies (*Figure 3*, g-l), but were also substantially stronger than *state-specific* ones (*Figure 3k*), indicating that *robust* responses are actively amplified within PCx.

## Pattern recovery depends on intracortical synaptic inputs

Piriform cortex is a recurrent cortical circuit that resembles auto-associative or discrete attractor networks. The ability to generate stable output patterns using degraded input is a property of such networks. If PCx is indeed such a network then the recurrent connectivity between PCx neurons should be essential for stabilizing odor representations across states. PCx contains two distinct classes of principal cells: pyramidal cells (PYRs), which are located primarily in deeper layer II and receive both OB and recurrent collateral inputs, and semilunar cells (SLs), which are more superficial and receive strong OB input but weak or no recurrent input (*Figure 4a*; *Suzuki and Bekkers, 2006*; *Choy et al., 2015*). To test whether recurrent connectivity predicts response stability, we generated a Netrin G1-Cre mouse line (*Ntng1*-Cre) that selectively expresses Cre recombinase in SLs in PCx (*Figure 4*, b and c). We then virally expressed either Cre-dependent Archaeorhodopsin-3 (Arch), Jaws, or

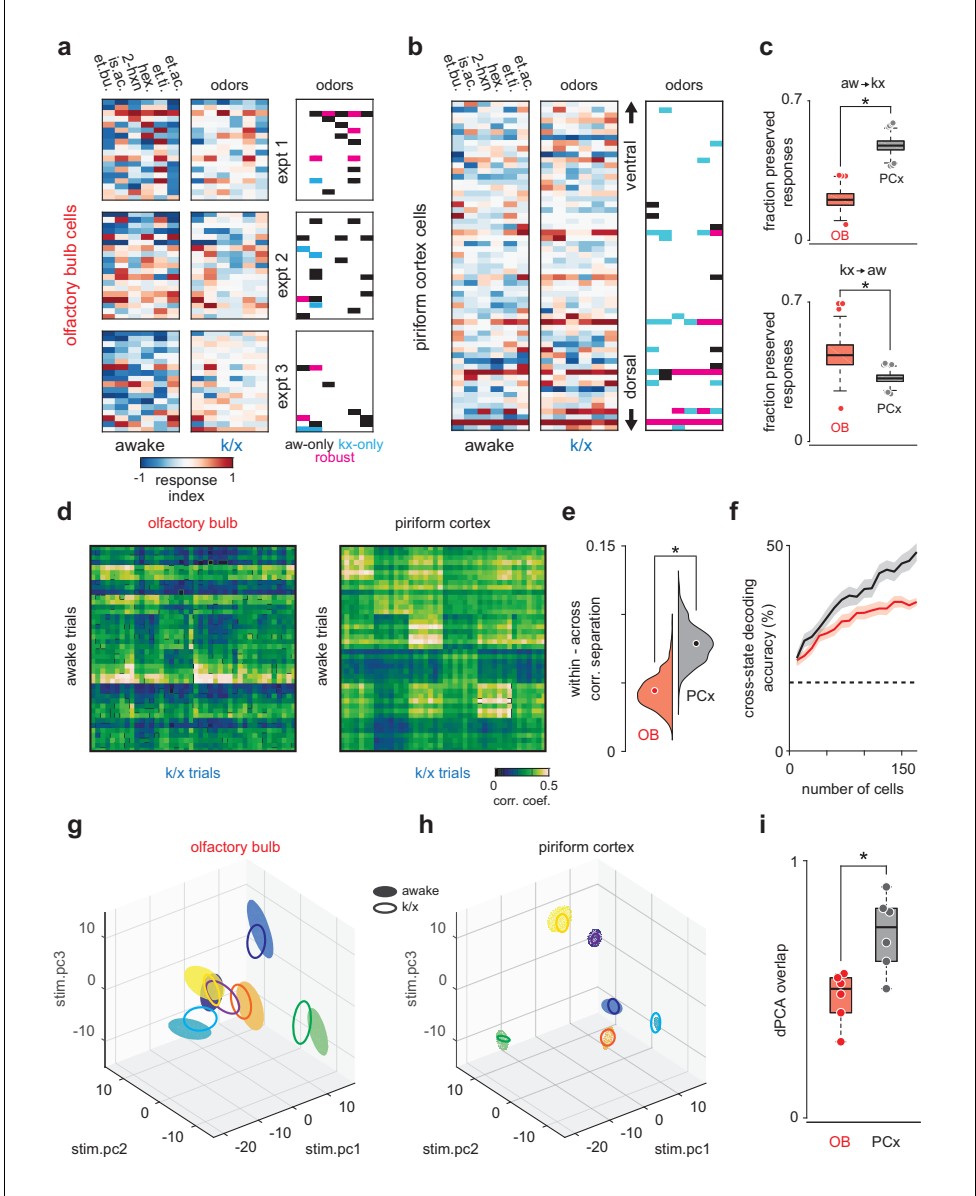

**Figure 2.** PCx stabilizes odor representations across activity regimes. (a–b) Example population responses to six odors for three OB (a) and one PCx recordings (b). The response index indicates the reliability of the difference between pre-odor and odor-evoked spiking (i.e. auROC*2–1). Awake-only (black), k/x-only (cyan), and robust cell-odor pairs (magenta), which were activated in both states ('activated' = p<0.05 rank-sum test), are indicated at right. PCx cells are sorted by their dorsal-ventral location. Note more robust PCx responses in deeper, (i.e. more dorsal) layer II. (c) *Top*, the fraction of significant awake cell-odor pair responses that are preserved under anesthesia in OB (n = 187 cells, 12 experiments) and PCx (n = 640 cells, 11 experiments). *Bottom*, the fraction of significant cell-odor pair responses under anesthesia that are observed in awake condition. Asterisks indicate p<0.05 on bootstrap difference test (aw to k/x: p<0.001; k/x to aw: p=0.023). (d) Cross-state trial-by-trial pseudopopulation correlation matrices sorted by odor within OB and PCx illustrate the similarity between awake and anesthetized population responses. The horizontal bands in these matrices indicate awake trials that had consistently high (light bands) or consistently low correlations (dark bands) with k/x responses regardless of odor. Typically, dark bands occur on awake trials with higher firing rates. Note that for these analyses, we used trials 6–13, and the regularity of these bands, especially in OB, reflects the progressive adaptation of responses, which are described in detail in *Figure 6*e. Separation (mean within-odor correlations - mean across-odor correlations) of cross-state odor representations in OB and PCx over 1000 bootstrap iterations. Asterisks indicate p<0.05 in bootstrap difference test (OB vs PCx, p<0.001). Means are shown as filled circles. (f) Cross-state decoding accuracy (trained on awake trials, tested on k/x trials) in OB and PCx. Mean ± 95% bootstrapped confidence intervals. (g) OB pseudopopulation activity projected onto the first three stimulus-dependent demixed PCA components and shown as mean ± 1 s.d. ellipsoids across trials of the same odor. Different colors correspond to different odors. Filled ellipsoids are awake responses, unfilled ellipsoids are anesthetized responses. (h) As in g, but for PCx. Note both the overall larger separation between odors in either state and the greater overlap of same-odor responses across states. (i) Overlap between awake and k/x trial response distributions projected onto their first three stimulus-dependent components in OB (red) and PCx (black). Dots show overlap scores for

*Figure 2 continued on next page*

Figure 2 continued

individual odors. Overlap is calculated using Matusita's overlap measure (see Materials and methods). Unpaired t-test, n = 6 odors, t(10) = −3.41, p=0.007.

The online version of this article includes the following figure supplement(s) for figure 2:

**Figure supplement 1.** State-dependent low-dimensional representations in OB and PCx.

Channelrhodopsin-2 (ChR2) in PCx to optogenetically differentiate SLs from PYRs during population recordings (*Figure 4*, d-f, *Figure 4—figure supplement 1*; 22.4 ± 16.0% opto-tagged, putative SL cells, n = 12 recordings). Indeed, odor responses were better-preserved across states in PYRs than SLs: (*Figure 4*, g and h). *Ntng1*-Cre mice were generally more sensitive to k/x anesthesia than the *Emx1*-Cre mice that we used in our control recordings, resulting in greater suppression of overall activity in *Ntng1*-Cre mice, and therefore a slightly lower fraction of total robust responses. SLs also

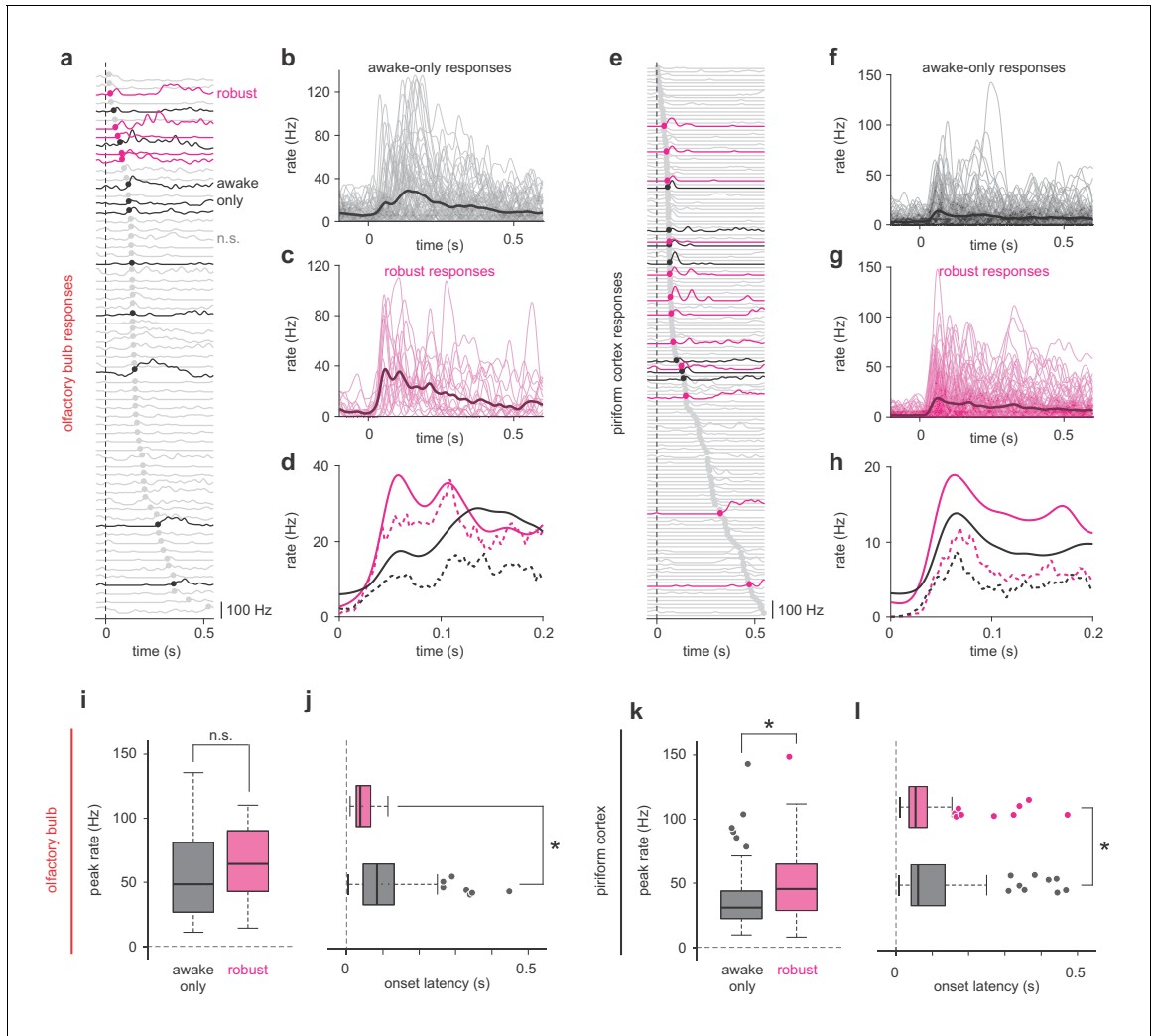

**Figure 3.** Robust PCx representations derive from short-latency OB responses. (a,e) All cell-odor pair responses for example simultaneously recorded OB (a) and PCx (e) populations sorted by onset latency determined by 2.5 st. dev. threshold crossing. Latencies are marked with filled circles. Robust, awake-only, and non-significant responses are magenta, black, and gray, respectively. (b–c) Overlay plots of all OB awake-only responses (b) or robust responses (c). The bold line is the mean response. (d) Mean (solid line) and median (dashed line) awake-only (black) and robust (magenta) OB responses. (f–h) As in b-d, but for PCx. (i–j) Peak firing rates (i) and onset latencies (j) for robust vs awake-only responses in OB. Boxes indicate quartiles and whiskers indicate ± 2.7 st. dev. from mean. Data points outside this range are shown as circles. Asterisks indicate p<0.05 in unpaired t-test. n = 81 awake-only OB cell-odor pairs and 21 robust OB cell-odor pairs. Peak: t(100) = −0.82, p=0.42; Latency: t(100) = 3.06, p=0.003. (k–l) As in i-j but for PCx. n = 125 awake-only PCx cells and 116 robust PCx cells. Peak: t(239) = −4.49, p=1.12e-5; Latency: t(236) = 2.47, p=0.01.

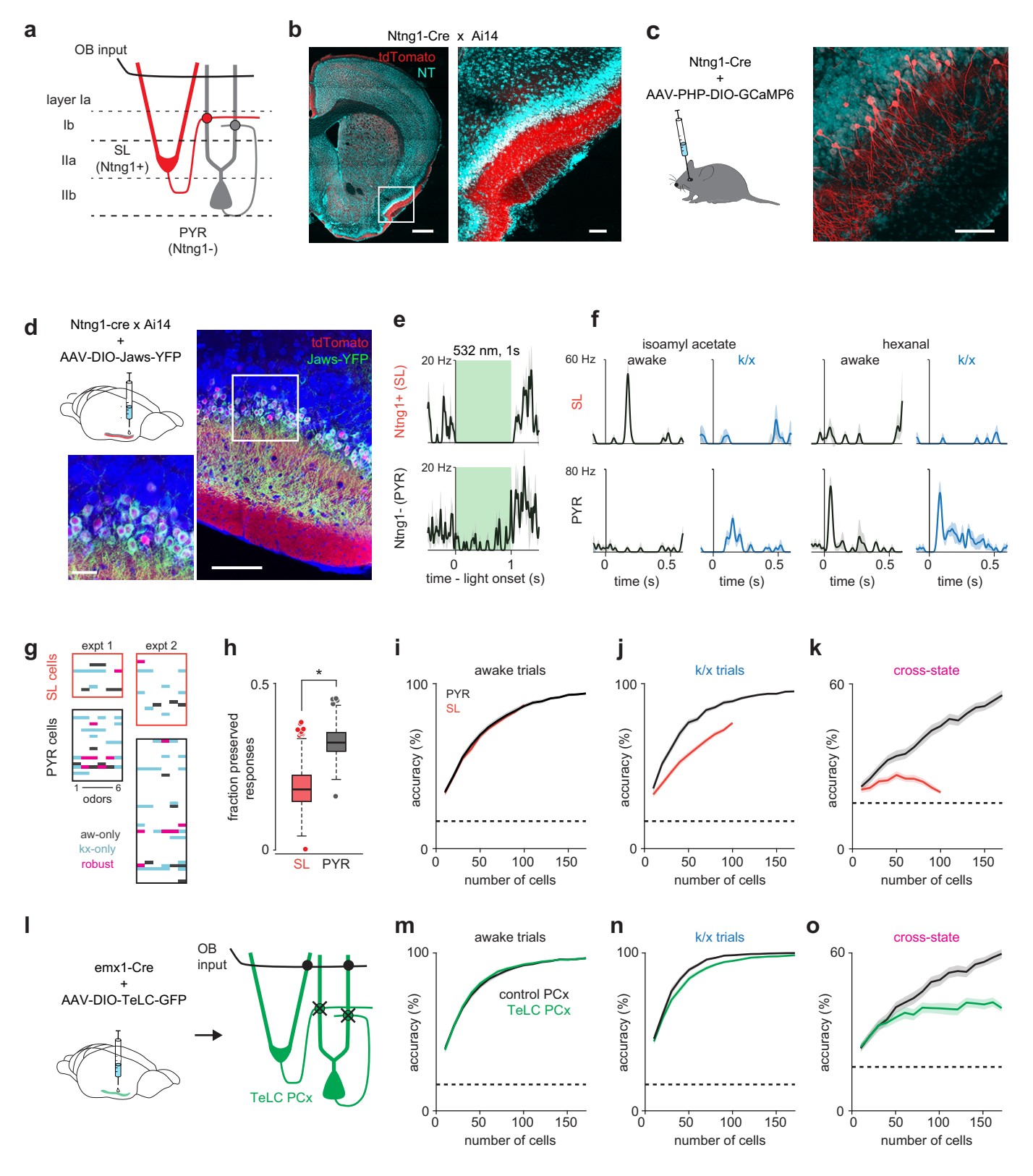

**Figure 4.** Pattern recovery requires recurrent circuits. (a) Schematic of inputs to excitatory cell-types in PCx. Semilunar cells only receive OB input; pyramidal cells receive OB and recurrent collateral inputs. (b–c) Selective expression in PCx semilunar cells using the *Ntng1*-Cre driver line. (b) Strong expression of Cre-dependent tdTomato in PCx layer IIa of *Ntng1*-Cre X Ai14 mice. Scale bars: 500 μm and 100 μm. (c) Sparse Cre-dependent GCaMP6

*Figure 4 continued on next page*

*Figure 4 continued*

expression shows *Ntng1+* cells restricted to superficial layer II (i.e. layer IIa) and lacking basal dendrites. Scale bar: 100 μm. (d) Identifying *Ntng1+* (semilunar, SL) and *Ntng1-* (pyramidal, PYR) cells in vivo using optogenetic inhibition. Injection of AAV expressing Cre-dependent Jaws in anterior piriform cortex restricts expression to cells with semilunar localization and morphologies. Scale bars: 100 μm and 20 μm. (e) Simultaneously recorded example cells exhibiting unambiguous suppression (top) or residual spiking (bottom) in response to 1 s, 532 nm light pulses. Traces show mean ± SEM responses over 40 laser pulses. Optogenetic tagging and unit stability criteria identified 108 SL and 234 PYR cells in 12 experiments from seven mice. (f) Odor responses during awake and k/x trials for example SL (top) and PYR cells (bottom). (g) *State-specific* (black, awake-only; cyan k/x only) and *robust* (magenta) responses in simultaneously recorded populations of SL and PYR cells from two example experiments. (h) The fraction of significant awake cell-odor pair responses that are preserved under anesthesia in SL and PYR. Asterisk indicates p<0.05 on bootstrap difference test (p=0.036). (i–j) Odor classification accuracy as a function of pseudopopulation size using SL (red) and PYR (black) cells in awake (i) and anesthetized (j) states. Mean ± 95% bootstrapped confidence intervals. (k) Cross-state decoding accuracy using SL (red) and PYR (black) cells. Mean ± 95% bootstrapped confidence intervals. (l) Strategy for disabling recurrent circuits in PCx. Expression of tetanus toxin in all PCx excitatory cells disrupts recurrent connectivity. (m–n) Odor classification accuracy as a function of pseudopopulation size in TeLC-infected (green) and contralateral control (black) PCx in awake (m) and anesthetized (n) states. Mean ± 95% bootstrapped confidence intervals. Pseudopopulations were built from 241 cells recorded in the control hemisphere in 4 experiments with 4 mice and 214 cells from the TeLC hemisphere in 6 experiments with five mice. (o) Cross-state decoding accuracy in TeLC- (green) and control (black) PCx. Mean ±95% bootstrapped confidence intervals.

The online version of this article includes the following figure supplement(s) for figure 4:

**Figure supplement 1.** Criteria for identifying opto-tagged Ntng1+ cells.
**Figure supplement 2.** Across-experiment variability in response preservation depends on preserved spontaneous activity.
**Figure supplement 3.** Decoding from TeLC-ipsilateral and contralateral OB populations.

recapitulated the degradation of OB representations under anesthesia (*Figure 4*, i and j) and the poor cross-state decoding (*Figure 4k*), whereas PYRs successfully recovered representations across states. Thus, two PCx cell populations, receiving the same feedforward input but distinguished by their recurrent connectivity, differentially recover odor representations when OB input is degraded.

Next, we asked whether eliminating recurrent connectivity in PCx abolished the ability to recover odor representations across states. To test this prediction, we injected conditional viral vectors to express tetanus toxin (AAV-DIO-GFP-TeLC) into PCx of *Emx1*-Cre mice to express TeLC in all PCx excitatory neurons (*Figure 4l*), effectively converting PCx into a pure feedforward circuit driven by OB (*Bolding and Franks, 2018*). We then obtained simultaneous bilateral recordings from TeLC-infected and contralateral control PCx hemispheres before and during anesthesia. Interestingly, odor responses could be classified equally accurately in awake responses from control and TeLC PCx (*Figure 4m*), and TeLC-PCx decoding was only slightly impaired under anesthesia (*Figure 4n*). Critically, however, cross-state decoding in PCx was markedly degraded in TeLC PCx (*Figure 4o*), indicating that recovery of the odor-specific features of the cortical response during anesthesia requires recurrent connections. To understand why within-state decoding was only minimally affected by TeLC expression, we examined decoding in OB ipsi- and contralateral to TeLC expression. Because TeLC expression blocks all PCx output from infected neurons, including strong projections from PCx to OB inhibitory neurons, OB output is substantially larger ipsilateral to TeLC-PCx (*Bolding and Franks, 2018*), and this may compensate for the loss of recurrent circuitry within PCx. Indeed, decoding using OB responses ipsilateral to TeLC-PCx was substantially better than in contralateral control OB in both awake and anesthetized conditions (*Figure 4—figure supplement 2a*). However, cross-state decoding in TeLC ipsilateral OB remained poor (*Figure 4—figure supplement 2b*), indicating that stronger ipsilateral TeLC-PCx OB responses can support reliable within-state decoding, but cannot fully rescue a state-dependent reconfiguration of OB odor representations.

## Timescale-dependent noise correlations in PCx populations

Our working hypothesis, that odor-selective PCx ensembles are recruited through recurrent connections, predicts some coordinated activity between simultaneously recorded PCx cell-pairs. However, previous work by Miura et al. reported near-zero noise correlations during PCx odor responses, appearing to preclude strong intracortical recruitment (*Miura et al., 2012*). Our data are consistent, both with these previous results and with the idea that an attractor network should show some weakly correlated activity across trials. Using Miura et al.'s calculation methods (in a 120 ms window following inhalation onset), noise correlations are indeed near-zero (*Figure 5*). However, correlated activity generated by an interconnected recurrent network should show correlations on timescales

more consistent with synaptic time constants. When we used smaller bin sizes, consistent with coordinated local activity, piriform cell-pairs do show weak noise-correlations just-after the peak of the odor-evoked response. In general, noise correlations are higher at all time points under anesthesia, reflecting low-frequency coordination (i.e. rhythmic spiking). Thus, at short timescales, noise-correlations can be observed in PCx during odor responses, consistent with the ability of PCx cells to recruit each other and retrieve stored odor-specific patterns.

## Rapid pattern formation in PCx populations

The enhanced stability of pattern activation across conditions in PCx is consistent with a pattern completion-like process occurring in PCx recurrent circuits. For stable population activity patterns to be retrieved they must first be formed; this is commonly thought to occur through experience. We therefore sought evidence for odor pattern formation in PCx in our recordings. If learning occurs during early trials then odor responses should systematically converge toward a stable pattern. We tested this prediction by measuring distances in neural activity space for population responses. Responses on initial trials were highly variable but then stabilized over subsequent trials (*Figure 6a–c*). Much of this variability can be explained by increased sniffing (*Figure 6d*), and therefore stronger overall responses, on early trials, however a multiple linear regression analysis revealed a significant effect of trial number alone (*Figure 6e*). Recording instabilities or changes in cell health did not account for this shift as corresponding changes did not occur for pre-trial activity (*Figure 6b*). Trial-by-trial stabilization curves were similar across odors and experiments (*Figure 6—figure supplement 1*). We considered the possibility that some part of this increased stability could be inherited from, and imparted to, OB. Our ability to interpret single-trial OB responses is limited by relatively low number of simultaneously recorded OB neurons, nevertheless we found that OB responses indeed stabilized over early trials (*Figure 6f–j*). OB stabilization dynamics were more rapid than in PCx, and there was a decrease in odor-evoked overall firing rates in PCx that was not apparent in OB. However, the contribution of centrifugal inputs from PCx to OB makes it impossible to determine whether the trial-dependent effects observed in OB reflects changes in OB circuitry that are then imparted to PCx or changes in PCx circuitry that are then imparted to OB.

To account for these effects, as well as receptor desensitization, sniffing, and arousal, we compared the stabilization of responses across trials in bilateral recordings from control versus TeLC-infected PCx in the same animals. Under these conditions, we again observed a significant trial-number effect in contralateral control hemispheres, but sniffing and overall rate fully accounted for response variability in TeLC-infected PCx, with no significant trial number effect (*Figure 6k and l*). These data support the hypothesis that learning is instantiated in PCx by recurrent circuitry in a trial-dependent manner.

## Short-term pattern stability in recurrently-connected PCx cells

Activity patterns within an attractor network should persist even after input is removed. To examine the temporal stability of PCx odor representations, we trained and tested a linear decoder on population responses (smoothed with a 200 ms kernel) from odor onset until several seconds after odor

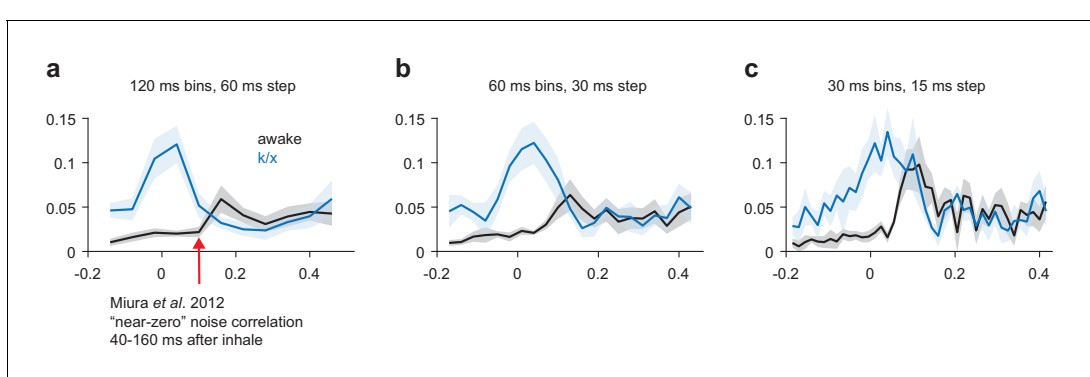

**Figure 5.** Noise correlations in awake and anesthetized PCx. (a–c) Average noise correlations in a sliding window surrounding odor inhalation for all awake (black) or k/x (blue) odor trials using a 120 ms (a), 60 ms (b) or 30 ms (c) bin size for spike counts. Mean ± s.e.m, n = 11 experiments.

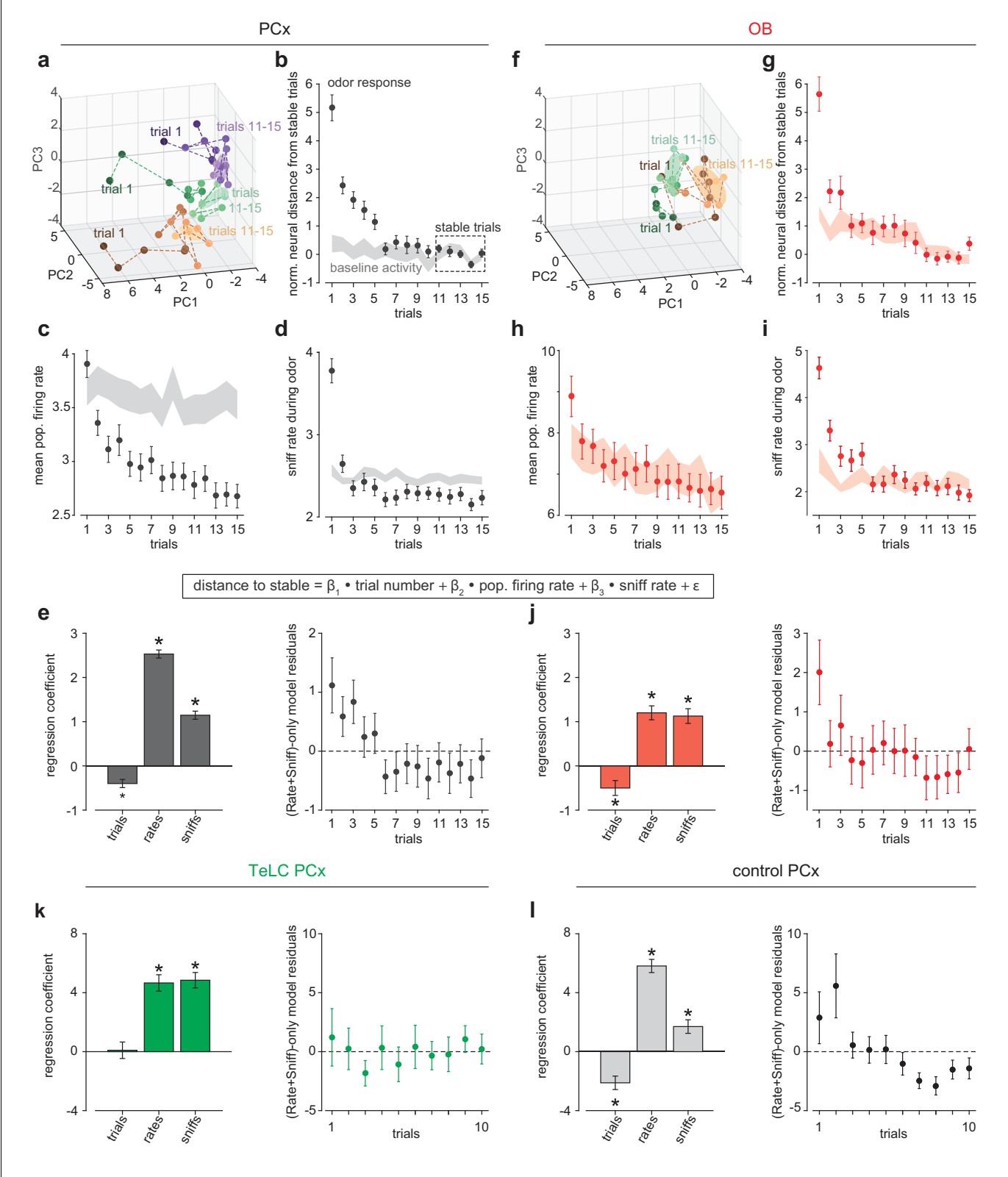

**Figure 6.** Rapid pattern formation in PCx population responses. (**a, f**) PCA trajectories for the first 15 presentations of three odors in an example simultaneous PCx (**a**) and OB (**f**) recording. The third odor in OB data occupied overlapping PC space and was omitted for visual clarity. The area occupied during the designated 'stable' trials is shown as mean ± 1 s.d. ellipsoids. Different colors correspond to different odors. (**b, g**) Average Euclidean distance from trial population vectors to stable trials normalized by the average distance between stable trials (b, PCx: n = 132 experiment-

*Figure 6 continued on next page*

odor pairs, g, OB: n = 45 experiment-odor pairs, mean ± SEM). The shaded area shows distances computed using pre-odor baseline activity (mean ± SEM). (c, h) Average population firing rates during odor response (mean ± SEM) or pre-odor baseline (shaded area, mean ± SEM) for PCx (c) or OB (h). (d, i) Average sniffing rates during odor response (mean ± SEM) or pre-odor baseline (shaded area, mean ± SEM) for PCx recordings (d) and OB recordings (i). (e) *Left,* multiple linear regression coefficients for effects of sniff rate, population firing rate, and trial number on population distance to stable in PCx (mean ± SEM). All main and interaction coefficients are significant (p<0.05). *Right,* Residuals plot of multiple linear regression on distance-to-stable fit with only sniff rate and population firing rate predictors, showing decreasing distance with trial number independent of these predictors. (j) As in e, but for OB recordings. (k, l) As in e, but for TeLC-infected PCx (k, n = 36 experiment-odor pairs) and contralateral control PCx (l, n = 24 experiment-odor pairs). Distance changes are fully explained by sniff rate and overall population firing rate in TeLC-PCx, but depend on trial number in control PCx.

The online version of this article includes the following figure supplement(s) for figure 6:

**Figure supplement 1.** Similar trial-trial population response stabilization across odors.

offset in responses from TeLC-infected and contralateral control hemispheres. In control PCx, odors could be accurately identified using activity at least 4 s after odor offset, but odor information

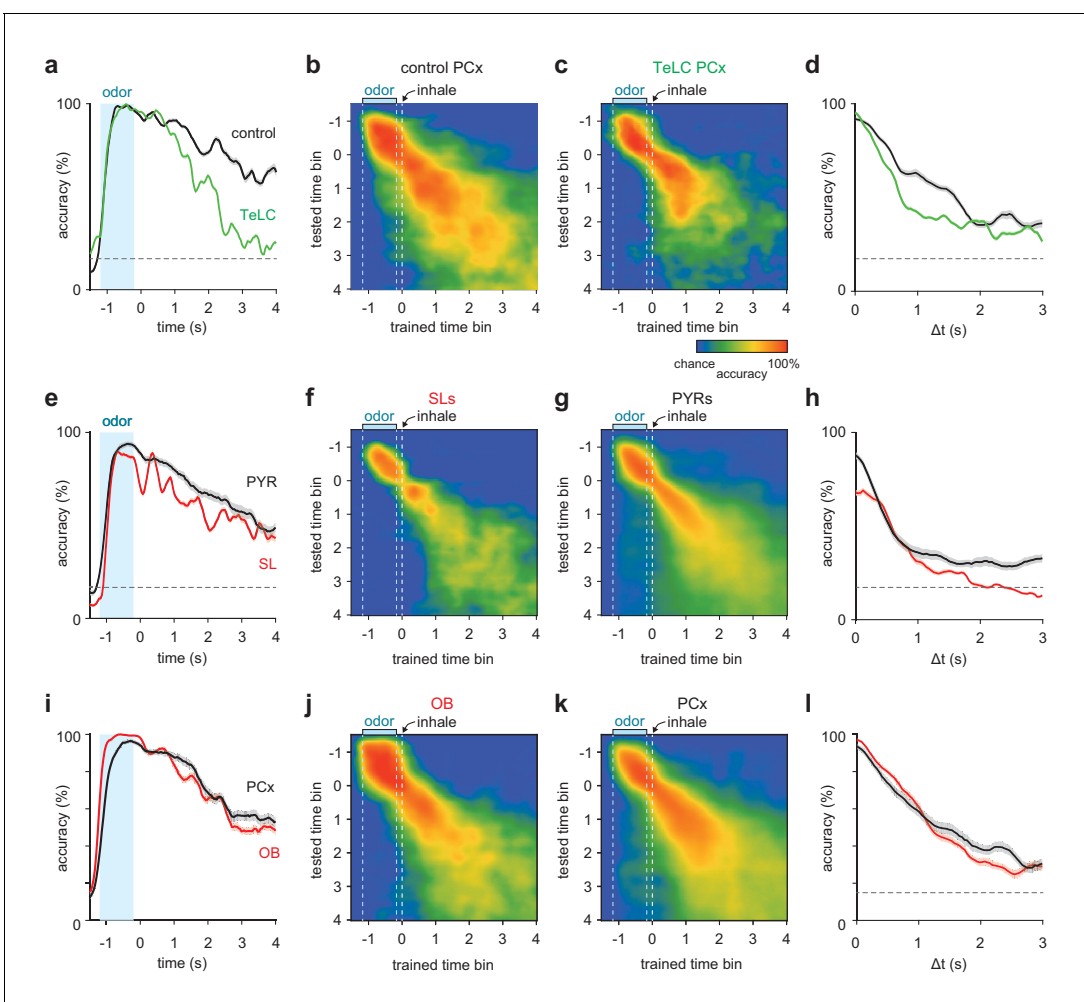

**Figure 7.** Short-term pattern stability in recurrently-connected PCx cells. (a–c) Average cross-time decoding accuracy for pseudopopulations of 200 cells recorded from control (b) or TeLC-infected (c) PCx. Responses were aligned to the first sniff after odor offset. (a) Decoding accuracy for training and testing on identical time bin at increasing times from odor offset for control (black) and TeLC- (green) PCx. Mean ± 95% bootstrapped confidence intervals. (d) Mean decoding accuracy at increasing temporal distance between the training bin (t = 0) and test bin in control (black) and TeLC- (green) PCx. These data are reflected in the rightmost dashed vertical lines in a and b. Mean ± 95% bootstrapped confidence intervals. (e–h) As in a-d, but for pseudopopulations of 100 SLs (f, red) or PYRs (g, black). (i–l) As in a-d, but for pseudopopulations of 170 OB cells (j, red) or PCx cells (k, black).

decayed much more rapidly after odor offset in TeLC-infected PCx (*Figure 7*, a-c). We then asked if odor representations remained stable at later time points or whether patterns of activity evolved dynamically across time (*Friedrich and Laurent, 2001*; *Bathellier et al., 2008*). To do this we trained our decoder on responses at odor offset and then tested on subsequent epochs. We could accurately classify responses long after odor offset in responses from control hemispheres, whereas decoding accuracy from TeLC-PCx decayed more rapidly (*Figure 7*, b-d). Similarly, odor-evoked activation patterns in SLs were less stable than those in simultaneously recorded PYRs in *Ntng1*-Cre mice (*Figure 7*, e-h). These results are consistent with a role for recurrent circuitry preserving odor representations in PCx after the stimulus has ended. However, we note that OB representations decayed only slightly more rapidly than in PCx (*Figure 7*, i-l), leaving open the possibility that OB-PCx interactions contribute to short-term maintenance of odor representations.

## Discussion

PCx activity exhibits several features that are consistent with attractor network function. First, cortical sensory representations are preserved even when upstream representations are substantially degraded. Second, odor responses stabilize over multiple presentations, reflecting, in part, the formation of cortical odor templates. Third, odor representations persist after the stimulus has ended. And, two lines of evidence indicate that recurrent cortical circuitry underlies these phenomena. First, responses destabilize and do not converge in a trial-number dependent manner after blocking transmitter release from PCx principal neurons. Second, odor responses are more stable in recurrently connected PYRs than in SLs.

### Limitations of chemogenetic approach for disabling recurrent circuits

Recurrent circuitry is classically defined as connections specifically between a single class of neurons – in this case, excitatory synaptic connections between pyramidal cells in PCx. We used the expression of TeLC in PCx excitatory cells to block synaptic output from principal cells (both SL and PYR) and thus eliminate recurrent circuity. However, in addition to blocking potential ensemble recruitment through excitatory-excitatory connections, this strategy also eliminates (1) cortical feedback projections primarily driving inhibitory granule cells in OB, (2) recruitment of feedback inhibitory circuits within PCx, and (3) any signaling to downstream targets. Pattern recovery may occur through coordinated activity in the PCx-OB loop which is disrupted by TeLC expression, but this is unlikely given that pattern recovery also degrades in SLs when the centrifugal PCx-OB connections are intact. Given that *robust* OB inputs occur earlier than *state-dependent* ones, cortical feedback inhibition likely enhances cross-state decoding by making PCx less responsive to later OB inputs, in the same way these circuits help maintain odor identity representations across odor concentrations (*Bolding and Franks, 2018*). However feedback inhibition cannot explain why *robust* PCx responses are significantly stronger than *state-dependent* ones; instead, this observation argues for an active amplification of this information by excitatory recurrent inputs. We cannot disprove that stabilization involves an elaborate loop from PCx to a downstream area and then back to PCx. Moving forward, recent technical developments that enable optogenetic control of the specific spatiotemporal patterns of OB output (*Chong et al., 2020*) could be used to examine pattern formation and recovery in PCx more directly.

### Stable odor representations in PCx

Even under normal, stable behavioral conditions, the olfactory system is challenged with identifying complex odors and odorant mixtures, with fluctuating odorant concentrations, and inherent variability at the level of receptor binding and odorant sensory neuron activity. Recurrent connections within PCx are capable of undergoing plasticity and forming self-reinforcing neural ensembles (*Kanter and Haberly, 1990*; *Jung et al., 1990*; *Poo and Isaacson, 2007*), which endows the circuit with the ability to form coherent sensory representations from noisy input (*Wilson and Sullivan, 2011*; *Haberly, 2001*). The destabilized and degraded OB odor responses that we induced by anesthesia throw the stabilizing function of PCx processing into stark relief, but we also see subtle evidence of this stabilizing function in greater odor separability in PCx over OB populations in the awake state (*Figure 1m*).

Previously, we described how long-range recurrent circuits recruit strong, global feedback inhibition, implementing a 'temporal winner-take-all' filter that allows the earliest-active OB inputs to largely define the PCx response and thereby make PCx odor responses robust to changes in odorant concentration (*Bolding and Franks, 2018*). This net-inhibitory process is complementary to the more constructive role for recurrent circuits that we propose here, in which recurrent connections help selectively recruit specific subsets of similarly responsive PCx principal cells.

We have described a trial-dependent stabilization of PCx odor representations, which we propose emerges through a Hebbian learning process at recurrent synapses between co-active PCx principal cells. Interestingly, (*Jacobson et al., 2018*) recently reported a trial-dependent shift in odor representations in the zebrafish homolog of PCx that appeared to require synaptic plasticity – that is, it was NMDA receptor-dependent – but their responses did not appear to stabilize over eight trials. Instead, our results are consistent with a recent study from *Pashkovski et al., 2020*, who showed that recurrent circuits transform OB inputs to form a systematic and somewhat-invariant map of chemical odor space that preserves correlational structure across odors, presumably by the same unsupervised learning process that we are proposing here. Thus, the stabilization we observe may reflect a default, activity-dependent sculpting that occurs in PCx during passive odor sensing. However, rodents can either be trained to either discriminate between or generalize across similar odors, depending on context (*Chapuis and Wilson, 2012*). The balance between generalization or discrimination may be modulated at multiple processing stages to serve changing behavioral contexts (*Chapuis and Wilson, 2012*; *Koldaeva et al., 2019*; *Chu et al., 2016*). Intriguingly, cholinergic (*Hasselmo and Bower, 1992*), noradrinergic (*Hasselmo et al., 1997*), and GABAergic (*Tang and Hasselmo, 1994*; *Franks and Isaacson, 2005*) modulators selectively affect recurrent but not afferent synapses in PCx, providing a mechanism to coherently and flexibly implement task-dependent computations in the same circuit.

## State-dependent odor responses in OB

We saw a decrease in OB odor responsivity after injection of k/x, which allowed us to examine how PCx responds with degraded input. However, previous studies have reported either no change (*Kollo et al., 2014*) or an increase in OB responses under anesthesia (*Kato et al., 2012*; *Rinberg et al., 2006*). Stimulus panels in each of these previous studies consisted primarily of neutral monomolecular odorants, similar to those we used, also diluted in the 0.1–10% range. We recorded from a similar population of ventrally located M/T cells and used an identical dose of ketamine/xylazine to *Rinberg et al., 2006*. The most substantive methodological differences between our study and theirs are 1) their freely-moving, awake animals were able to actively truncate the odor presentation by removing their nose from the odor port, whereas in our experiments head-fixed animals passively received odors for a full second and 2) their odor responses were aligned to nose withdrawal rather than to inhalation. These would both have the effect of decreasing the apparent strength of awake responses and may largely account for the differences in our results.

It is more difficult to compare the results from our extracellular recordings with calcium imaging results from *Kato et al., 2012*. We made additional OB recordings matching the length of their odor stimulus (4 s), accounting for the smoothing and loss of individual spikes associated with calcium imaging, and using a range of ketamine doses from 70 to 200 mg/kg, however we were unable to identify conditions in which we observed larger and/or more reliable odor responses in anesthetized OB (data not shown).

We also compared the changes we observed in spontaneous and odor-evoked activity to those reported by *Kollo et al., 2014* using in vivo whole-cell recordings. By contrast with their intracellular data, we saw an overall decrease in firing rather than a narrowing of firing rate distributions in anesthetized OB activity. These differences may be partially explained by the large population of so-called 'silent' mitral cells, which we likely underrepresent with extracellular spike-sorting procedures. Nevertheless, we found that even very low firing rate cells in our recordings tend to decrease their firing under anesthesia and do not appear more likely to have strong odor responses. Future experiments will be required to resolve these discrepancies. However, regardless of why our results are different, they do not impact our fundamental observation that PCx odor representations remain robust when OB responses are degraded.

## Odor representations in semilunar and pyramidal cells

Our findings indicate functional segregation between the coding properties of SLs and PYRs. Recent studies have demonstrated that local connectivity, molecular identity, and projection targets vary with depth in PCx. Superficial/SL cells receive input from OB and target PCx PYRs as well as more posterior targets such as lateral entorhinal cortex and cortical amygdala (*Diodato et al., 2016*; *Mazo et al., 2017*). Because SLs receive little or no input from other PCx excitatory neurons, their responses are minimally affected by local cortical computations (*Choy et al., 2015*; *Suzuki and Bekkers, 2011*). These cells may thus simply integrate converging inputs from OB with no mechanisms to correct response variability inherited from OB. As such, the downstream targets of SLs may receive more variable, state-dependent olfactory representations.

Hippocampal area CA3 has been modeled as an auto-associative network capable of storing and retrieving a large number of unique representations due to the presence of an extensive network of recurrent excitatory collaterals (*Treves and Rolls, 1994*). Similarly, because PCx PYRs receive inputs from other local PCx excitatory neurons, they could retrieve previously-stored odor-evoked activity patterns via recurrent reactivation (*Johnson et al., 2000*; *Franks et al., 2011*). We propose the initial activation of a small subset of superficial PCx neurons can drive reactivation of a stable ensemble of deeper cells, enabling the recovery of PCx odor responses despite degraded OB input. Interestingly, PCx cells in deep layer II project back to OB (*Diodato et al., 2016*; *Mazo et al., 2017*; *Luskin and Price, 1983*), suggesting that an accurate representation of the current stimulus is returned to the OB, allowing comparison between ongoing input and the retrieved activation patterns. This loop allows SLs to receive OB input that is updated by feedback from PYRs, and may explain why odor representations in SLs are better preserved after odor offset than those in TeLC-PCx (*Figure 6*).

We hypothesize that PCx circuits undergo rapid plasticity induced by stimulus exposure which embeds attractor states in the PCx synaptic architecture and later biases the trajectories of PCx population activity to previously visited states. We have demonstrated that these processes require recurrent excitation in PCx but the detailed mechanisms remain to be explored. The bulk of experience-driven changes we observe in PCx are accompanied by decreases in overall stimulus-evoked spiking, suggesting that plasticity in both excitatory and inhibitory connectivity may effect response stabilization (*Vogels et al., 2011*; *D'amour and Froemke, 2015*; *Frank et al., 2019*). Also, it remains unclear whether the instability in TeLC-PCx is due to the inability to retrieve patterns or to store them in the first place. Addressing these questions will require the development of temporally restricted and synapse-type specific interventions. Nevertheless, our results demonstrate that computations carried out by recurrently connected cells within PCx enable pattern stabilization and strongly suggest that PCx serves as a locus of memory storage for odor representations acquired through incidental sensory experience.

## Materials and methods

All experimental protocols were approved by Duke University Institutional Animal Care and Use Committee. The methods for head-fixation, data acquisition, electrode placement, stimulus delivery, and analysis of single-unit and population odor responses are adapted from those described in detail previously (*Bolding and Franks, 2017*). A portion of the data reported here (5 of 12 simultaneous OB-PCx experiments) were also described in that previous report.

### Mice

All mice except *Ntng1*-cre mice were adult (>P60, 20–24 g) offspring of Emx1-Cre (+/+) breeding pairs obtained from The Jackson Laboratory (005628).

Ntng1-Cre knock-in mouse was generated using CRISPR based method in which sequences encoding the Cre recombinase followed by the bovine growth hormone polyA sequences were inserted at the start codon ATG of Ntng1 gene. During CRISPR mediated homologous recombination, an extra 162 base pairs was also inserted in front of the ATG start codon of Cre (note that these 162 base pairs are in-frame with Cre). The 162 extra DNA sequences are: atgtatttgtcaagattcctgtcgatccatgccctgtgggtgacagtgtcctctgtgatgcagccctaccttacattatcagatctgaattcactagtcgcgcccggggagcccaaaggttaccccagttggggcgggcccgaacgaaaaggtagggctgcc .

Thus, the resulting allele contains Cre with an extra 5' end 34 amino acids inserted into the start codon of the Ntng1 gene. The knock-in allele was verified using both genomic PCR and Southern blot.

## Head-fixation

Mice were habituated to head-fixation and tube restraint for 15–30 min on each of the two days prior to experiments. The head post was held in place by two clamps attached to ThorLabs posts. A hinged 50 ml Falcon tube on top of a heating pad (FHC) supported and restrained the body in the head-fixed apparatus.

## Ketamine/xylazine anesthesia

To induce anesthesia, *Emx1*-Cre animals were injected with a bolus ketamine/xylazine cocktail (100/10 mg/kg, ip). This induced stable anesthesia lasting 30–45 min. In initial experiments, many Ntgn1-cre mice died shortly after administering this highly standardized k/x dose, and although dosage was decreased 10–20% in subsequent experiments, these mice appeared 'more deeply anesthetized'. Throughout the recording, body temperature was maintained using a heating pad (FHC). During anesthesia breathing became metronomic and animals ceased spontaneous forelimb movements.

We defined a subset of 7 awake trials and 7 k/x trials for all analyses except the analysis of odor representation stabilization in *Figure 5A–E*. Awake trials were selected as the last seven trials prior to k/x injection, to minimize sniff-related variability which was prevalent in early trials. K/X trials were defined as the seven trials following behavioral and electrophysiological onset of anesthesia effects, most prominently expressed in the loss of variable breathing frequency and increase in low-frequency LFP power.

## Data acquisition

Electrophysiological signals were acquired with 32-site polytrode acute probes (A1 × 32-Poly3-5mm-25s-177, Neuronexus) through an A32-OM32 adaptor (Neuronexus) connected to a Cereplex digital headstage (Blackrock Microsystems). Unfiltered signals were digitized at 30 kHz at the headstage and recorded by a Cerebus multichannel data acquisition system (BlackRock Microsystems). Experimental events and respiration signal were acquired at 2 kHz by analog inputs of the Cerebus system. Respiration was monitored with a microbridge mass airflow sensor (Honeywell AWM3300V) positioned directly opposite the animal's nose. Negative airflow corresponds to inhalation and negative changes in the voltage of the sensor output.

## Electrode placement

The recording probe was positioned in the anterior piriform cortex using a Patchstar Micromanipulator (Scientifica). For piriform cortex recordings, the probe was positioned at 1.32 mm anterior and 3.8 mm lateral from bregma. Recordings were targeted 3.5–4 mm ventral from the brain surface at this position with adjustment according to the local field potential (LFP) and spiking activity monitored online. Electrode sites on the polytrode span 275 μm along the dorsal-ventral axis. The probe was lowered until a band of intense spiking activity covering 30–40% of electrode sites near the correct ventral coordinate was observed, reflecting the densely packed layer II of piriform cortex. For simultaneous ipsilateral olfactory bulb recordings, a micromanipulator holding the recording probe was set to a 10-degree angle in the coronal plane, targeting the ventrolateral mitral cell layer. The probe was initially positioned above the center of the olfactory bulb (4.85 AP, 0.6 ML) and then lowered along this angle through the dorsal mitral cell and granule layers until encountering a dense band of high-frequency activity signifying the targeted mitral cell layer, typically between 1.5 and 2.5 mm from the bulb surface.

## Spike sorting and waveform characteristics

Individual units were isolated using Spyking-Circus (https://github.com/spyking-circus) (*Yger, 2018*; *Yger et al., 2018*). Clusters with >1% of ISIs violating the refractory period (<2 ms) or appearing otherwise contaminated were manually removed from the dataset. Pairs of units with similar waveforms and coordinated refractory periods in the cross-correlogram were combined into

single clusters. Unit position with respect to electrode sites was characterized as the average of all electrode site positions weighted by the wave amplitude on each electrode. Relative dorsal-ventral unit position was determined by fitting the waveform positions within a recording with a truncated normal distribution (truncated at 0 and 275) and then subtracting the mean of this fit.

## Unit stability criteria

To assure stable identification of cells across states, sorted units were subjected to further stability criteria (*Figure 1—figure supplement 2*). Units were excluded if their overall rate fell below 0.01 Hz, if their rate changed more than 100-fold across states, or if their average peak-to-peak waveform amplitude changed by more than 50 µV across states. 107/294 OB units and 63/703 PCx units were discarded based on these criteria.

## Percent responding and sparseness bootstrap analyses

Cell-odor pair responsivity was visualized using a response index (2*auROC-1) comparing the distribution of odor-evoked spike counts to the pre-odor baseline. Cell-odor pairs were labeled as significantly odor-responsive using a rank-sum test (p<0.05) on trial-by-trial spike counts over the first sniff after odor delivery compared to spike counts over the last pre-odor sniff. Because of variability in cell yield for olfactory bulb experiments we computed population responsivity measures on the population of all recorded cells rather than for each experiment. We established confidence bounds for these measures (percent activated responses, lifetime and population sparseness) by sampling these with replacement from the population of all recorded cells 1000 times. For significance testing, a null distribution of mean differences (between awake and k/x samples) was constructed by randomly selecting equivalent-sized samples from the combined population 1000 times, and the p-value was the fraction of null responses more extreme than the empirical mean difference.

## Trial-trial population vector correlations

The similarity of population odor responses, defined as spike count vectors within the first sniff, was quantified using the Pearson correlation coefficient. Population responses were combined across experiments to form a pseudopopulation, and correlations for all trial-pairs were calculated (i.e. seven trials for two odors = 49 correlations). The correlation between two stimuli (across odor) or between a stimulus and itself (within odor) was then taken as the average of these correlations. Separation of population firing rate vectors in neural space was the average within-odor correlation minus the average across-odor correlation. Bootstrap significance testing on pseudo-population correlation measures were determined with the null hypothesis that mean differences between OB and PCx could be generated from a homogenous population containing cells from both regions. We combined OB and PCx responses and sampled this distribution with replacement to match the recorded population size of OB and PCx cells and then computed the mean difference between correlation separation for these populations 1000 times. The p-value was then the fraction of null differences that were more extreme than the empirical mean difference.

## Noise correlations

Noise correlations were defined as the correlation in across-trial variability around the mean stimulus-evoked response between simultaneously recorded cell-pairs and were calculated according to the methods described in *Miura et al., 2012*. Briefly, spike count responses within a specified temporal window around inhalation were z-scored across trials for a given stimulus. Then, correlation coefficients were computed across trials for each stimulus and for each cell-pair in each simultaneously recorded population. These values were first averaged within an experiment across cell-pairs and stimuli and then averaged across experiments.

## Population decoding analysis

Odor classification accuracy based on population responses was measured using a linear multi-class SVM classifier with 10-fold cross-validation (LIBLINEAR, solver 4 (Crammer and Singer method), https://www.csie.ntu.edu.tw/~cjlin/liblinear/ (*Fan et al., 2008*). Responses to six distinct monomolecular odorants presented at 0.3% v/v were used as the training and testing data. The feature vectors for spike count classification were the spike counts for each cell during the 480 ms following

inhalation. For decoding across states, the classifier was trained on all awake trials and accuracy was assessed across all anesthetized trials.

Classification accuracy was measured across multiple pseudopopulation sizes. To estimate the mean accuracy at each size, we constructed pseudopopulations by randomly subsampling from the entire recorded population 200 times. Bootstrap confidence intervals on the mean accuracy for each population size were estimated by sampling with replacement from the distribution of accuracies and re-computing the mean 1000 times.

### Local field potential analysis

30 kHz raw recordings were downsampled to 1 kHz and filtered between 0.05–500 Hz with a 3-pole Butterworth filter. Average power spectra and OB-PCx coherence were obtained using the multi-taper spectrum utilities in the Chronux package (www.chronux.org). For visualization, LFPs were pre-whitened with an 2nd-order autoregressive filter, and spectrograms were computed in a 30 s sliding window in 3 s steps.

### Spontaneous activity and pairwise phase consistency

Spontaneous activity was computed using spikes that occurred >3 s after odor offset and >2 s before odor onset. The relationship of each unit's spiking to the ongoing respiratory oscillation was quantified using pairwise phase consistency (PPC) as in *Vinck et al., 2010*. Each spike was assigned a phase by interpolation between inhalation (0 degrees) and exhalation (180 degrees). Each spike was then treated as a unit vector and PPC was taken as the average of the dot products of all pairs of spikes.

### Demixed PCA overlap analysis

Demixed PCA projections were computed using the Machens lab, MATLAB implementation (https://github.com/machenslab/dPCA) (*Kobak et al., 2016b*; *Kobak et al., 2016a*). Demixed PCA finds separate reduced-rank regression solutions that best reconstruct population responses averaged over each experimenter-defined factor. This is conceptually similar but not identical to performing PCA on, for instance, the stimulus-averaged population response separately from a PCA performed on state-averaged population responses. Trial-by-trial pseudopopulation responses were constructed as above from all recorded cells using a 500 ms response window. PCs were then computed on Stimulus, State, and Stimulus X State interaction marginalizations. Trial responses were projected onto the top 3 Stimulus components and the trial-mean and covariance of these 3D projections for each odor in each state were determined. These measures were then used to compute an overlap score for each odor across states according to Matusita's measure (*Minami and Shimizu, 1999*; *Matusita, 1966*).

$$\xi = \frac{2^{\frac{n}{2}}|\Sigma_1|^{\frac{1}{4}}|\Sigma_2|^{\frac{1}{4}}}{|\Sigma_1 + \Sigma_2|^{\frac{1}{2}}} exp\left\{-\frac{1}{4}(\mu_1 - \mu_2)^T(\Sigma_1 + \Sigma_2)^{-1}(\mu_1 - \mu_2)\right\}$$

### Optogenetic tagging of *Ntng1*[+] cells

In 5/12 experiments with *Ntng1*-Cre mice, putative semilunar cells were tagged by Cre-dependent expression of Jaws (a red-shifted variant of halorhodopsin), and in 5/12 experiments with *Ntng1*-Cre mice, putative semilunar cells were tagged by Cre-dependent expression of Archaeorhodopsin (Arch). Cells expressing either inhibitory opsin show rapid onset and deep suppression during exposure to 532 nm (Arch) or 640 nm (Jaws) light. Normal odor response recordings were made with an optic fiber attached probe, and 20 1 s laser pulses were delivered at the end of the experiment. Two criteria were applied to identify Arch[+] cells: 1) p<0.0001 in rank-sum test of spiking in the 1 s preceding and during laser stimulation, 2) median last-spike latency during laser pulse <0.01 ms (shown as red dots). Additionally, cells with overall firing rates < 0.175 Hz or a peak-trough time <0.35 ms in their average waveform were excluded from classification either as Arch[+] or Arch[-] cells.

2/12 experiments used excitation with channelrhodopsin for opto-tagging. Cells were stimulated using ~200, 1 ms pulses of 473 nm light, delivered at 4 Hz at the end of the experiment. Two criteria were applied to identify ChR2[+] cells: 1) p-value<0.001 in Stimulus-Associated spike Latency Test (*Kvitsiani et al., 2013*), 2) latency-to-peak response in PSTH <0.003 ms. Cells with a peak-trough time <0.35 in their average waveform were excluded from classification as ChR2[+] or ChR2[-].

### Neural distance-to-stable and regression analysis

For each experiment, population firing rate vectors for each odor trial were constructed from spiking during the 500 ms after odor inhalation or from a control period 2000 to 1500 ms before odor inhalation. Mean Euclidean distance from each trial population response to the last five awake odor responses (stable trials) was normalized to the mean Euclidean distance between stable trials to estimate population distance-to-stable. For the main analysis of PCx and OB population trajectories, PCx data were combined from simultaneous OB-PCx recordings and *Ntng1*-Cre recordings, and three OB-PCx experiments which had <15 awake trials prior to k/x injection were omitted (n = 22 PCx recordings; n = 9 OB recordings).

To examine the influence of sniffing, overall firing rates, and odor experience over trials on the stabilization of population responses, we fit a multiple linear regression model using the *fitlm* function in the MATLAB Statistics and Machine Learning toolbox with sniff rate, firing rate, and trials, as predictors of distance-to-stable. Sniff rate was estimated as the reciprocal of the first breath duration following odor presentation. Population firing rates were taken as the mean response across neurons on each trial. All predictors were z-scored to allow comparison of the magnitude of regression coefficients. To further visualize stabilization across trials that is independent of sniffing and overall firing rates, we fit a reduced model including only sniff rate and firing rate as predictors, and examined the residuals of this model as a function of trials.

### Cross-time decoding and temporal stability

To assess stability of odor representations across short timescales, we trained and tested an SVM classifier (as above) on different time bins following odor offset. Smooth pseudopopulation responses were built from kernel density functions (200 ms kernel) aligned to the first inhalation after odor offset, and the classifier was trained and tested on each combination of time points up to 1.5 s before and 4 s after inhalation. Classification accuracy was assessed with leave-one-out cross-validation and no time bins from the test trial were included in the training data for each fold.

## Acknowledgements

We thank J Beck and G Field for helpful discussions and A Fleischmann, L Glickfeld, S Lisberger and A Schaefer for helpful comments on earlier versions of the manuscript. This work was supported by grants from NIDCD (DC015525 and DC016782) and the Edward Mallinckrodt Jr. Foundation.

## Additional information

### Funding

| Funder | Grant reference number | Author |
| --- | --- | --- |
| National Institute on Deafness and Other Communication Disorders | DC015525 | Kevin M Franks |
| National Institute on Deafness and Other Communication Disorders | DC016782 | Kevin M Franks |
| National Institute of Neurological Disorders and Stroke | U19 NS112953 | Kevin M Franks |
| National Institute of Neurological Disorders and Stroke | NS 077986 | Fan Wang |

The funders had no role in study design, data collection and interpretation, or the decision to submit the work for publication.

### Author contributions

Kevin A Bolding, Conceptualization, Data curation, Software, Formal analysis, Validation, Investigation, Visualization, Methodology; Shivathmihai Nagappan, Data curation, Investigation, Visualization; Bao-Xia Han, Fan Wang, Resources; Kevin M Franks, Conceptualization, Resources, Data curation,

Formal analysis, Supervision, Funding acquisition, Validation, Investigation, Visualization, Methodology, Project administration

### Author ORCIDs
Kevin A Bolding ⦿ https://orcid.org/0000-0002-2271-5280
Kevin M Franks ⦿ https://orcid.org/0000-0002-6386-9518

### Ethics
Animal experimentation: This study was performed in strict accordance with the recommendations in the Guide for the Care and Use of Laboratory Animals of the National Institutes of Health. All experimental protocols were approved by Duke University Institutional Animal Care and Use Committee (protocols A177-18-07). All surgeries were performed under either ketamine/xylazine and/or isoflurane anesthesia, and every effort was made to minimize suffering.

### Decision letter and Author response
Decision letter https://doi.org/10.7554/eLife.53125.sa1
Author response https://doi.org/10.7554/eLife.53125.sa2

## Additional files

### Supplementary files
- Transparent reporting form

### Data availability
Raw data and code are available on Dryad (https://doi.org/10.5061/dryad.n2z34tmtj) and GitHub (https://github.com/FranksLab/eLife2020-recurrents-stabilize; copy archived at https://github.com/elifesciences-publications/eLife2020-recurrents-stabilize), respectively.

The following dataset was generated:

| Author(s) | Year | Dataset title | Dataset URL | Database and Identifier |
|---|---|---|---|---|
| Bolding KA, Nagappan S, Han BX, Wang F, Franks KM | 2020 | Data from: Recurrent circuitry is required to stabilize piriform cortex odor representations across brain states | https://doi.org/10.5061/dryad.n2z34tmtj | Dryad Digital Repository, 10.5061/dryad.n2z34tmtj |

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
