## [Decision Letter]

**Acceptance summary:**

This study reveals a role for recurrent neural circuits in stabilizing odor representations in the olfactory cortex. The authors observe that anesthesia degrades odor representations in the olfactory bulb but not in the olfactory cortex. They then use genetic approaches to perturb synaptic function and reveal a role for recurrent circuits in maintaining odor-evoked cortical responses. The study advances mechanistic understanding of networks involved in olfactory processing. The study is also likely to be of interest to others who do not study olfaction but are generally interested in recurrent networks.

**Decision letter after peer review:**

Thank you for submitting your article "Recurrent circuitry is required to stabilize piriform cortex odor representations across brain states" for consideration by *eLife*. Your article has been reviewed by Laura Colgin as the Senior Editor, a Reviewing Editor, and three reviewers. The reviewers have opted to remain anonymous.

The reviewers have discussed the reviews with one another and the Reviewing Editor has drafted this decision to help you prepare a revised submission.

The reviewers thought the conclusions of the manuscript were potentially exciting, but that further analysis of existing data was required to validate key claims. The two key points of debate concerned (1) the validity of claims related to the attractor network and recurrent circuitry and (2) the potentially gross effects of anesthesia. Essential revisions related to these general points, and other major and minor points, are detailed below.

Summary:

The authors demonstrate that while anesthesia disrupts olfactory bulb (OB) representations of odors profoundly, the representations of odors between wakefulness and anesthesia remain relatively stable in the piriform cortex (PCx). Also, silencing output of PCx excitatory neurons reduced separability of odor representations in PCx. Odor responses in PCx were initially unstable but approached a stable pattern after repeated odor stimulation. The authors conclude that these features of the PCx resemble aspects of self-stabilizing / attractor type of networks and are a result of the recurrent circuitry in PCx, given that odor representations of optogenetically identified semilunar cells in PCx (cells that lack recurrent collaterals) degrade under anesthesia, whereas odor representations of optogenetically identified pyramidal cells in PCx (cells that receive collateral inputs) are largely preserved across states. Reviewers felt that the use of anaesthetized vs. awake states as the key comparison for degraded vs. preserved OB input was a weakness, and that it would be very compelling in the future to explicitly provide degraded input (e.g. optogenetically or through odor mixtures missing components). However, they felt that the results were still compelling, and they agreed that such experiments were well beyond the scope of the present study. Suggestions for future experiments that were provided by the reviewers are mentioned below for the authors' information, although they were ultimately not considered to be Essential Revisions.

Essential revisions:

1) One major concern is that the changes in odor-induced activity underlying the observed effects of anesthesia on correlations and separability of odor representations are not described in detail. A more detailed analysis of these effects is important to better understand how anesthesia changes odor representations in the OB, and why odor representations in PCx are not affected in the same way. Several specific comments related to this general point are provided below:

a) Reviewers were surprised by the poor odor representation in the anaesthetized OB, given results of prior studies. There is a long list of papers demonstrating seemingly robust, odor dependent responses in the OB of anaesthetized rodents (e.g., Yokoi et al., 1999, Dhawale, 2010, Buonviso et al., 1992, admittedly none of them directly demonstrating decoding ability). For a long time, the general conclusion was that waking up mice and rats resulted in MC activity becoming less odor- and more context-sensitive, less robust etc (e.g. RInberg et al., 2006, Kay and Laurent, 1999). While a lot of the apparent lack of reliable coding in the awake state was traced back to variability in sniff recording, the puzzle remains as to why Bolding et al. see such poor odor representation in the anaesthetized OB that even an ensemble of 150 (putative Mitral and Tufted) cells does not allow for efficient classification of a panel of 6 quite different monomolecular odors (Figure 1N). Can the authors pinpoint why they observe such poor odor representation in the OB? Are they using e.g. much lower concentration than previous studies? Is it the nature of the odor panel? As the lack of accurate representation in the OB is the basis for their use of the anaesthetized input to PCX as a proxy for a corrupted input, this observation deserves more explanation / exploration. In other words, the authors come to very different conclusions about the relationship between awake and anesthetized activity in the OB than what has been presented in the literature. The authors discuss this at length in subsection “State-dependent odor responsivity”. Nevertheless, this is a puzzling and possibly important point and more emphasis in the abstract and Results section might be needed to highlight this appropriately – and possibly help to resolve contradicting conclusions of published papers. Importantly, the statement in subsection “Odor responsivity is state-dependent in OB but not PCx” that their findings are consistent with previous studies is misleading. The authors need to explicitly discuss these inconsistencies at that point in the text.

b) It is important to better understand the changes in odor responses that cause high correlations and low separability in the OB under k/x anesthesia. One possibility is that there is a pattern of odor-independent "background activity" from a subset of cells that respond to all (most) odors. The horizontal "band-like" structure in the OB correlation matrix in Figure 2D hints at this. It is also possible that tuning curves of all cells become more similar. The origin of high correlations need to be analyzed in more detail; this is important to understand the effects of anesthesia and the difference between brain states, and to interpret effects in piriform cortex.

c) The authors should also analyze correlations in spontaneous activity across populations and how they change under anesthesia. Does anesthesia induce a pattern of spontaneous activity that persists during odor responses?

d) It is possible that anesthesia has little effect on the odor response itself but induces a global activity pattern that is additive to the odor response and present in all odor responses (and perhaps also in spontaneous activity). This possibility is quite different from the possibility that odor responses change globally and become more overlapping. The authors should try to distinguish between these (and other possibilities) and consider the consequences for the interpretation of k/x effects in PCx.

e) How do excitatory and inhibitory responses change under anesthesia? Are both affected equally? How often did responses change character (from excitatory to inhibitory or vice versa)? What was the correlation between responses of single neurons to odors (tuning curves) in the awake and anesthetized state?

f) The mean spontaneous firing rates in the OB and PCx in awake state and during k/x should be reported explicitly in the manuscript. Effects of k/x on spontaneous firing should be compared to those observed previously, e.g. by Rinberg et al., 2006.

2) General points were raised regarding the authors' claims regarding attractor networks and recurrent circuitry. Specific points related to this are provided below.

a) Do the authors actually show that it is "recurrent circuitry" (implying recurrent excitation as in most models of attractor networks)? All their work is certainly consistent with recurrent excitation underlying the observed features of an attractor network. They show that SL cells – that are thought to lack input from SL or PYR – do not show the same "pattern completion" stability features and that blocking the output of pyramidal cells similarly blocks "pattern completion". Both are consistent with recurrent excitation in PCX playing a key role – however, it is also consistent with simply "needing piriform cortical PYR input" (recurrent or feedback or feedforward) or e.g. a key role for FBI (as in Bolding and Franks, 2018, maybe technically also recurrency, albeit not in the sense it is usually used in the description of pattern completion and attractor networks). This needs to be actively discussed, and the Abstract and Discussion section should be phrased more carefully to better reflect results (e.g., "Recurrent connections are required…" "We find that PCx is an attractor network by virtue of its recurrent activity"). For example, in the first sentence of Discussion section: this is too strong a statement. It is possible that the underlying observations also involve other brain areas. PCx shows signatures of an attractor network, but it is not fully resolved whether it is an attractor network in the classical mechanistic sense.

b) The adjustment of sniff frequency (i.e. the initial rapid sniffing mentioned in subsection “Odor responsivity is state-dependent in OB but not PCx” and excluded for analysis) should be an excellent example of "altered/corrupted input to PCX". How does PCX deal with such altered OB inputs? Wouldn't one expect some kind of stabilization acting here as well?

c) Stabilization is an important phenomenon that is somewhat underrepresented in the manuscript. It would be interesting to see some further analyses of this phenomenon (e.g. is stabilization observed for all odors? Does the time course of stabilization (number of trials) depend on the odor? How does it depend on inter-trial interval?).

d) Correlations in PCx increase globally under anesthesia; the effect looks in first approximation as if a global "background correlation" is added to the pattern of correlations in the awake state. How can a "background correlations" arise in an attractor network? This is not consistent with the classical view of canonical attractor networks and should be discussed. Can the increased background correlation be explained by the observed changes in activity in the OB?

3) The results and their interpretation by the authors suggest that PCx neurons should show correlated variability of activity across trials. This is inconsistent with results reported previously by Miura et al., 2012, who report low noise correlations. The authors should perform a similar analysis of noise correlations and compare their results to those of Miura et al. The open comparison of present results to those of previous studies would be appreciated in the Discussion section, even when there are discrepancies. The authors may consider including a comparison to Miura et al., here; this paper is currently not discussed in any detail.

4 Figure 4H: the fraction of preserved responses in both types of neurons is lower than the fraction of preserved responses pooled over all neurons in Figure 2C. Is this due to inter-animal variation? Please explain.

5) How well can odors be decoded from OB activity after stimulus offset? The analysis in Figure 6 should also be performed for OB responses.

6) It would be useful to see not only peak firing rates and onset of robust vs non-robust responses but also their full time courses. The authors could for example plot an overlay of all robust responses and their mean, same for non-robust responses.

7) Figure 2A,B show that there are fewer robust cells in the OB, but there are also fewer non-robust responses (black and blue). This figure suggests that there are more cell-odor-pairs responding to odors in PCx than in the OB, which is at odds with Figure 1F-I. Please explain. What is the response index? Please define. The fraction of preserved responses should be reported both ways (wrt awake responses and wrt to k/x responses), and wrt responses in either state.

8) The argumentation why OB output is not degraded under anesthesia when TeLC is expressed ipsilaterally is not consistent with the simplified interpretation that PCx representations do not degrade under anesthesia (in normal animals not expressing TeLC). In normal animals, the back-projection to the OB does not appear to degrade representations in the OB (results from awake animals). Then, removing back-projections should not improve representations in TeLC animals, unless there is a complex interaction between back-projections and anesthesia that is not understood.

9) The usage of "degraded" is somewhat unclear. It may be useful to use a more precise description here because "degraded" may be expected to mean "more noisy", which is expected to decrease correlations. This semantic issue is linked to the question of whether the increase in correlation can be explained by a decrease in noise. This should be addressed by an analysis of noise and intertrial variation of responses.

[Editors' note: further revisions were suggested prior to acceptance, as described below.]

Thank you for resubmitting your article "Recurrent circuitry is required to stabilize piriform cortex odor representations across brain states" for consideration by *eLife*. Your article has been reviewed by Laura Colgin as the Senior Editor, a Reviewing Editor, and three reviewers. The reviewers have opted to remain anonymous.

The reviewers have discussed the reviews with one another and the Reviewing Editor has drafted this decision to help you prepare a revised submission.

The reviewers felt that the revised manuscript was substantially improved, and after discussion, there was one point in particular that reviewers felt was not properly addressed. Many of the conclusions related to attractor network behavior assume, implicitly, that anesthesia "degrades" odor discrimination because it decreases signal to noise, but the data do not seem to support this view. An alternative possibility is that anesthesia results in some common mode of activity such that activity becomes less odor-specific. This should be addressable by some simple analysis, and discussed appropriately in a few sentences.

Reviewer comments are provided in full below in case they are helpful.

Reviewer #1:

The authors have responded to the comments. Overall, I feel that with the current changes the paper does provide reasonable support for its claims.

1) Anesthesia and correlations/separability of odor representations.

The authors do a detailed comparison with prior work and better place their observations in context with, and in contrast to, what has previously been seen.

1b). Poor odor representation in anaesthetized OB.

Again, the authors now discuss the prior work and acknowledge where there are differences.

2) Attractor networks.

The authors have put in further discussions and tempered their interpretations.

They have not addressed some other suggestions about testing for attractor behavior in the network based on sharp transitions between response states triggered by removal of components from a multi-component odor, and similar manipulations. As this is substantial work, it is fair enough to defer this for future study.

3) Noise correlations:

The authors do show that their results on long (120 ms) windows match earlier work from Miura, but point out that shorter windows exhibit some (weak) correlations.

Reviewer #2:

In the revision Bolding and Franks discuss all major points raised. They can't really resolve the key issues but lay them open rather transparently and discuss them clearly. Regarding the discrepancy of their recordings under anesthesia from the published literature they discuss potential reasons for the discrepancy and compare their results with the literature in detail. The very first statement they make is still somewhat misleading and it should rather be pointed out that their results differ from the literature upfront (l 51: "Anesthesia induced pronounced changes in respiration patterns, oscillatory activity and spontaneous spiking in both OB and PCx (Figure 1B and C; Figure 1—figure supplement 1), consistent with previous work ").

Subsection “Odor responsivity is state-dependent in OB but not PCx” are somewhat clumsy and confusing – they emphasize that reliable responses can be seen in their data, yet decoding is substantially impaired. This needs rephrasing pointing out that reliable responses are somewhat rare in their data compared to the published literature.

Reviewer #3:

The authors have addressed some but not all comments.

Comments:

1) The most important issue is that the authors have not elucidated how anesthesia "degraded" odor representations in the OB. They observe that across-trial correlations go up, which tells us that a decrease in signal or an increase in noise cannot easily account for the effect of anesthesia. Their explanation that "degraded" decoding is due to reduced responses is inconsistent with the observation that all correlations increase (if activity is weakened and the signal to noise ratio decreases, correlations should converge to zero. If signal to noise ratio increases, which is possible when anesthesia reduces baseline activity, the difference between within and across odor correlations should increase, not decrease). An obvious possibility is that odor stimulation evokes a non-specific activity pattern under anesthesia that is independent of odor identity. This needs to be tested, as was put forward under main point 1 in the previous review. The authors have addressed this point incompletely because they only tested whether a background pattern present during spontaneous activity could increase correlations during odor responses (by subtracting mean spontaneous activity), but they did not test whether a non-specific "background" pattern is evoked by odor stimulation. There are many obvious ways to test this possibility (identifying a common mode in patterns by PCA or so and subtracting it out, or asking whether it is always the same cells that are responsible for different across-odor correlations, etc). As pointed out before, this issue is important because understanding the effects in the OB – which are obviously not a simple noise increase – is essential to understand the effects in PCx. Most predictions related to attractor networks consider inputs with some form of "random" noise, but anesthesia does not seem to induce "random" noise.

2) Along the same lines: Subtracting background rates did have a clear effect on accuracy (probably significant; please test) that is somewhat understated in the manuscript. It increases with the number of cells, as expected for a "background pattern". Is this effect enhanced during odor stimulation?

3) Along the same lines: the horizontal bands in Figure 2D, left panel, have not been explained. Please test whether they come from a background pattern (that may be observed only during odor stimulation, or enhanced by odor stimulation). If not, what else could it be?

4) The components in activity patterns responsible for increased across-odor correlations in piriform cortex under anesthesia have not been identified. These across-odor correlations are not predicted by attractor networks. They seem to argue against attractor networks. The authors should try to identify the sources for these correlations and address in the Discussion section whether or not they are consistent with an attractor network hypothesis.

5) The question whether the gradual stabilization of odor responses is due to plasticity of responses in the OB or PCx can be addressed in TeLC-PC mice: if the site of pasticity is PCx, stabilization of responses in the OB should be absent. Figure 6K seems to support this possibility but showing more data for TeLC-PC mice (as in Figure 6G) could clarify this question further.

6) In *Emx1*-Cre mice, decoding seems to be even improved under k/x as compared to awake (compare Figure 4N to 4M), both in control and in TeLC expressing mice. Please explain.

7) The conclusion that robust responses are actively amplified within PCx appears to strong because it is based only on a rather crude analysis of peak firing rates.

8) It would be interesting to see correlation matrices such as Figure 1K that are generated only from SL or PYR cells.

---

## [Author Response]

Summary:The authors demonstrate that while anesthesia disrupts olfactory bulb (OB) representations of odors profoundly, the representations of odors between wakefulness and anesthesia remain relatively stable in the piriform cortex (PCx). Also, silencing output of PCx excitatory neurons reduced separability of odor representations in PCx. Odor responses in PCx were initially unstable but approached a stable pattern after repeated odor stimulation. The authors conclude that these features of the PCx resemble aspects of self-stabilizing / attractor type of networks and are a result of the recurrent circuitry in PCx, given that odor representations of optogenetically identified semilunar cells in PCx (cells that lack recurrent collaterals) degrade under anesthesia, whereas odor representations of optogenetically identified pyramidal cells in PCx (cells that receive collateral inputs) are largely preserved across states. Reviewers felt that the use of anaesthetized vs. awake states as the key comparison for degraded vs. preserved OB input was a weakness, and that it would be very compelling in the future to explicitly provide degraded input (e.g. optogenetically or through odor mixtures missing components). However, they felt that the results were still compelling, and they agreed that such experiments were well beyond the scope of the present study. Suggestions for future experiments that were provided by the reviewers are mentioned below for the authors' information, although they were ultimately not considered to be Essential Revisions.Essential revisions:1) One major concern is that the changes in odor-induced activity underlying the observed effects of anesthesia on correlations and separability of odor representations are not described in detail. A more detailed analysis of these effects is important to better understand how anesthesia changes odor representations in the OB, and why odor representations in PCx are not affected in the same way. Several specific comments related to this general point are provided below:

In response to this important point we carried out additional analyses on changes in OB responses across states to further examine the origin of degraded OB odor responses and representations under anesthesia.

1) Reviewers suggested that a strong ‘background’ activity pattern could dominate anesthetized activity, making it difficult to resolve distinct odor-evoked responses. To examine this, we looked at spontaneous activity correlations (sniff-sniff population vector correlations for activity during the pre-odor baseline). Results were similar to the across-odor correlationsin Figure 1L. That is, activity in both OB and PCx is more correlated across sniffs under anesthesia (Figure 1—figure supplement 4A). If increased background correlations explain poor separation of odor representations, we would expect an equal or even greater decrement in PCx separation and decoding. The fact that background correlations increase in both regions, but the decreased response separation is only observed in OB suggest that background correlations are not the primary explanatory factor.

2) Nevertheless, to further examine the effect of background activity on OB response classification, we found the mean rate during baseline for all cells and subtracted that from the odor responses before classifying, thus eliminating any consistent ‘background pattern’. OB decoding is only very slightly improved by this procedure (Figure 1—figure supplement 4B).

3) A naïve prediction of the background activity hypothesis is that baseline activity increases under anesthesia, especially in OB cells that lose their responses. However, we cannot find evidence for this in our data. Instead, our data suggest that OB cells lose their responses under anesthesia because their odor-evoked firing rates decrease (Figure 1—figure supplement 4B and 4B). We therefore think that OB decoding accuracy is poor under anesthesia because there are fewer/weaker OB responses.

4) OB decoding could also fail because responses become more variable under anesthesia. However, on average across all OB responses, the Fano factors are nearly the same in awake and k/x trials (awake: 2.72 ± 2.11, k/x: 2.43 ± 1.27; Figure 1—figure supplement 4E). We further broke this analysis down to examine Fano factors for awake-only or robust responses. There is some increased variability in the awake-only responses that could contribute to poor decoding, but maintained responses are even more reliable (Figure 1—figure supplement 4F). The bigger effect and probably greatest contributor to the decoding issue is simply the weaker and sparser responses under anesthesia.

a) Reviewers were surprised by the poor odor representation in the anaesthetized OB, given results of prior studies. There is a long list of papers demonstrating seemingly robust, odor dependent responses in the OB of anaesthetized rodents (e.g., Yokoi et al., 1999, Dhawale, 2010, Buonviso et al., 1992, admittedly none of them directly demonstrating decoding ability). For a long time, the general conclusion was that waking up mice and rats resulted in MC activity becoming less odor- and more context-sensitive, less robust etc (e.g. RInberg et al., 2006, Kay and Laurent, 1999).

We were also initially surprised by this result, given the existing literature. Nevertheless, we reliably observed that the anesthetic cocktail we injected reduced both responsivity and classification accuracy for responses in olfactory bulb. We wish to emphasize that our results, showing degraded OB responses and decoding in ketamine-xylazine (k/x) anesthetized mice should not be interpreted as a general statement about the effects of different anesthetics in different systems or even about k/x anesthesia effects across anesthesia protocols. Thus, we are not comfortable making direct comparisons to Yokoi et al., 1995, who used urethane anesthesia in rabbits; to Dhawale et al., 2010 who used a 40% lower dose than ours but sustained administration of the cocktail for >10 hours; or to Buonviso et al., 1992, where equitesin was used in anesthetized rats. We can indeed observe reliable odor responses amongst a portion of the mitral/tufted cells in these recordings (~5% of cell-odor pairs) and agree that it is possible to do so in a variety of anesthetized preparations. We have revised the text describing our observed anesthesia effects to clarify that we do not interpret this as a general result reflecting all forms of anesthesia.

While a lot of the apparent lack of reliable coding in the awake state was traced back to variability in sniff recording, the puzzle remains as to why Bolding et al. see such poor odor representation in the anaesthetized OB that even an ensemble of 150 (putative Mitral and Tufted) cells does not allow for efficient classification of a panel of 6 quite different monomolecular odors (Figure 1N). Can the authors pinpoint why they observe such poor odor representation in the OB? Are they using e.g. much lower concentration than previous studies? Is it the nature of the odor panel? As the lack of accurate representation in the OB is the basis for their use of the anaesthetized input to PCX as a proxy for a corrupted input, this observation deserves more explanation / exploration. In other words, the authors come to very different conclusions about the relationship between awake and anesthetized activity in the OB than what has been presented in the literature. The authors discuss this at length in subsection “State-dependent odor responsivity”.

As noted by the reviewer, we address this discrepancy and our efforts to explain the difference with studies directly comparable to ours (Kato et al., 2012, Rinberg et al., 2006, Kollo et al., 2014) in the Discussion section. We have added a concise description of odor stimuli across these studies. The choice of stimuli is unlikely to explain the differences we observed. Absent a direct comparison of results obtained from these diverse activity monitoring techniques in the same experimental preparation, we cannot confidently ascribe the differences in our anesthesia results to a specific experimental factor beyond the factors we proposed in the Discussion section.

Nevertheless, this is a puzzling and possibly important point and more emphasis in the abstract and Results section might be needed to highlight this appropriately – and possibly help to resolve contradicting conclusions of published papers. Importantly, the statement in subsection “Odor responsivity is state-dependent in OB but not PCx” that their findings are consistent with previous studies is misleading. The authors need to explicitly discuss these inconsistencies at that point in the text.

The point is well-taken and we have modified the text accordingly.

b) It is important to better understand the changes in odor responses that cause high correlations and low separability in the OB under k/x anesthesia. One possibility is that there is a pattern of odor-independent "background activity" from a subset of cells that respond to all (most) odors. The horizontal "band-like" structure in the OB correlation matrix in Figure 2D hints at this. It is also possible that tuning curves of all cells become more similar. The origin of high correlations need to be analyzed in more detail; this is important to understand the effects of anesthesia and the difference between brain states, and to interpret effects in piriform cortex.

See above (response to Essential revision 1).

c) The authors should also analyze correlations in spontaneous activity across populations and how they change under anesthesia. Does anesthesia induce a pattern of spontaneous activity that persists during odor responses?

See above (response to Essential revision 1).

d) It is possible that anesthesia has little effect on the odor response itself but induces a global activity pattern that is additive to the odor response and present in all odor responses (and perhaps also in spontaneous activity). This possibility is quite different from the possibility that odor responses change globally and become more overlapping. The authors should try to distinguish between these (and other possibilities) and consider the consequences for the interpretation of k/x effects in PCx.

See above (response to Essential revision 1).

e) How do excitatory and inhibitory responses change under anesthesia? Are both affected equally? How often did responses change character (from excitatory to inhibitory or vice versa)? What was the correlation between responses of single neurons to odors (tuning curves) in the awake and anesthetized state?

We have now included a description of changes in significant suppressed odor responses in OB and PCx under anesthesia in Figure 1(F and H), and include statistics for response sign-switching and tuning curve correlations in the main text.

f) The mean spontaneous firing rates in the OB and PCx in awake state and during k/x should be reported explicitly in the manuscript. Effects of k/x on spontaneous firing should be compared to those observed previously, e.g. by Rinberg et al., 2006.

We now include these measures in the Results section and compare our results to those from previous studies.

2) General points were raised regarding the authors' claims regarding attractor networks and recurrent circuitry. Specific points related to this are provided below.a) Do the authors actually show that it is "recurrent circuitry" (implying recurrent excitation as in most models of attractor networks)? All their work is certainly consistent with recurrent excitation underlying the observed features of an attractor network. They show that SL cells – that are thought to lack input from SL or PYR – do not show the same "pattern completion" stability features and that blocking the output of pyramidal cells similarly blocks "pattern completion". Both are consistent with recurrent excitation in PCX playing a key role – however, it is also consistent with simply "needing piriform cortical PYR input" (recurrent or feedback or feedforward) or e.g. a key role for FBI (as in Bolding and Franks, 2018, maybe technically also recurrency, albeit not in the sense it is usually used in the description of pattern completion and attractor networks). This needs to be actively discussed, and the Abstract and Discussion section should be phrased more carefully to better reflect results (e.g., "Recurrent connections are required…" "We find that PCx is an attractor network by virtue of its recurrent activity"). For example, in the first sentence of Discussion section: this is too strong a statement. It is possible that the underlying observations also involve other brain areas. PCx shows signatures of an attractor network, but it is not fully resolved whether it is an attractor network in the classical mechanistic sense.

The reviewer is saying that our results are consistent with recurrent circuitry being required for attractor dynamics and pattern completion, but also consistent with the simpler idea that piriform simply “needs PYR input”. Given that, by far, the biggest distinguishing feature of SLs vs. PYRs is that SLs are driven almost exclusively by OB input and PYRs are driven by both direct OB input and input from other SLs and PYRs, we think these two statements are practically, if not formally, equivalent. Moreover, our observation that *robust* and *awake-only* OB responses have similar magnitudes, but *robust* responses are stronger than *awake-only* responses in PCx indicates that these responses are actively amplified in PCx, which is difficult to explain as a function solely of recurrent feedback excitation, as in Bolding and Franks, 2018. We therefore think that we are justified is stating that our results show that recurrent circuitry is required to support these phenomena. Nevertheless, we do take the reviewers’ point that more baroque explanations, such as loops from PCx to other brain areas and back again, could also be involved and we have added a section in the Discussion section that describes the limitations in the interpretation, especially of the TeLC expression experiments. We have also tempered some the particularly strongly worded phrases.

b) The adjustment of sniff frequency (i.e. the initial rapid sniffing mentioned in subsection “Odor responsivity is state-dependent in OB but not PCx” and excluded for analysis) should be an excellent example of "altered/corrupted input to PCX". How does PCX deal with such altered OB inputs? Wouldn't one expect some kind of stabilization acting here as well?

In principle, an analysis of sniff frequency could be useful. However, this analysis is confounded in these experiments because, in addition to having very few (~3-5) variable sniff trials, variable sniffing occurs primarily in the earliest trials, when we predict the initial formation of the odor template occurs. In our working model, this ‘learned’ representation is the basis for pattern stabilization. We therefore predict that odor responses would not be especially stable in these early trials, before the pattern is ‘learned’.

c) Stabilization is an important phenomenon that is somewhat underrepresented in the manuscript. It would be interesting to see some further analyses of this phenomenon (e.g. is stabilization observed for all odors? Does the time course of stabilization (number of trials) depend on the odor? How does it depend on inter-trial interval?).

We now show stabilization traces for all population-odor pairs and show that the phenomenon is roughly equivalent with all odors. Inter-trial interval was held stable in the current experiments. We agree that examining the dependence on inter-trial interval would be interesting, and varying this parameter will be an important element of future experiments, but is beyond the scope of this study.

d) Correlations in PCx increase globally under anesthesia; the effect looks in first approximation as if a global "background correlation" is added to the pattern of correlations in the awake state. How can a "background correlations" arise in an attractor network? This is not consistent with the classical view of canonical attractor networks and should be discussed. Can the increased background correlation be explained by the observed changes in activity in the OB?

See above (response to Essential revision 1).

3) The results and their interpretation by the authors suggest that PCx neurons should show correlated variability of activity across trials. This is inconsistent with results reported previously by Miura et al., 2012, who report low noise correlations. The authors should perform a similar analysis of noise correlations and compare their results to those of Miura et al. The open comparison of present results to those of previous studies would be appreciated in the Discussion section, even when there are discrepancies. The authors may consider including a comparison to Miura et al., here; this paper is currently not discussed in any detail.

We have now performed an extensive analysis of noise correlations and describe the outcomes of this analysis in the Results section (Figure 5). This is a valuable addition to our paper and we thank the reviewers for this suggestion. As we now describe in the text, our data are consistent, both with Miura and with the idea that an attractor network should shows some weak correlated activity across trials. We find near-zero noise correlation when using a 120 ms time window, as Miura et al., did, but we do find weak noise correlations when using time windows more in line with synaptic time constants. This warrants a more thorough investigation because there are numerous factors that can contribute to noise correlations, including spiking rate and oscillatory activity, but that is beyond the scope of this paper.

4 Figure 4H: the fraction of preserved responses in both types of neurons is lower than the fraction of preserved responses pooled over all neurons in Figure 2C. Is this due to inter-animal variation? Please explain.

Yes, we noticed this too. This difference almost certainly depends on animal strain, as we observed that *Ntng1*-Cre mice were considerably more sensitive to k/x. We have elaborated on this observation in Figure 4—figure supplement 2, which shows that the rate of response preservation can be partially predicted by the maintenance of spontaneous activity and that overall rates are more sensitive to anesthesia in a portion of *Ntng1*-Cre recordings. We consider the comparisons of simultaneously recorded cell populations in the same genetic background to be most reliable.

5) How well can odors be decoded from OB activity after stimulus offset? The analysis in Figure 6 should also be performed for OB responses.

We have included this analysis in (now) Figure 7.

6) It would be useful to see not only peak firing rates and onset of robust vs non-robust responses but also their full time courses. The authors could for example plot an overlay of all robust responses and their mean, same for non-robust responses.

We have included the suggested illustration in Figure 3.

7) Figure 2A,B show that there are fewer robust cells in the OB, but there are also fewer non-robust responses (black and blue). This figure suggests that there are more cell-odor-pairs responding to odors in PCx than in the OB, which is at odds with Figure 1F-I. Please explain. What is the response index? Please define. The fraction of preserved responses should be reported both ways (wrt awake responses and wrt to k/x responses), and wrt responses in either state.

We thank the reviewers for pointing out this inconsistency. These examples were generated from an older version of analysis code that included the earliest awake trials and longer periods of anesthesia, neither of which could be described as a steady-state. Our final analysis was limited to late awake trials after odor-evoked sniffing had subsided and a shorter, stable period of anesthesia that excluded early trials after induction and later trials during which recovery may add to response variability. Examples and summary data are now generated using the same procedure and we have replaced the examples in Figure 2A and B with more representative data. We now define the response index in the methods. We now also report preserved responses both ways, though we argue the awake-to-k/x preservation is the relevant measure for our analysis.

8) The argumentation why OB output is not degraded under anesthesia when TeLC is expressed ipsilaterally is not consistent with the simplified interpretation that PCx representations do not degrade under anesthesia (in normal animals not expressing TeLC). In normal animals, the back-projection to the OB does not appear to degrade representations in the OB (results from awake animals). Then, removing back-projections should not improve representations in TeLC animals, unless there is a complex interaction between back-projections and anesthesia that is not understood.

Our previous presentation was unnecessarily confusing and our explanation was garbled, and we apologize for this. We have simplified the presentation of the OB decoding figure with TeLC (Figure 4—figure supplement 3A) to show OB decoding accuracy in awake vs. anesthetized states in OBs ipsi- and contralateral to TeLC on the same graph. As can now be seen clearly, although OB decoding is substantially better ipsilateral to TeLC-PCx in both awake and anesthetized conditions, OB decoding does decrease under anesthesia in both cases. Note however that ipsilateral OB decoding under anesthesia is still actually slightly better than decoding in awake contralateral control OB. We think that this stronger OB output improves PCX decoding and helps compensate for the loss of recurrent circuitry in both awake and anesthetized conditions.

9) The usage of "degraded" is somewhat unclear. It may be useful to use a more precise description here because "degraded" may be expected to mean "more noisy", which is expected to decrease correlations. This semantic issue is linked to the question of whether the increase in correlation can be explained by a decrease in noise. This should be addressed by an analysis of noise and intertrial variation of responses.

We have revised the text to clarify our usage of this term, and we include an analysis of response variability in Figure 1—figure supplement 4.

[Editors' note: further revisions were suggested prior to acceptance, as described below.]

The reviewers felt that the revised manuscript was substantially improved, and after discussion, there was one point in particular that reviewers felt was not properly addressed. Many of the conclusions related to attractor network behavior assume, implicitly, that anesthesia "degrades" odor discrimination because it decreases signal to noise, but the data do not seem to support this view. An alternative possibility is that anesthesia results in some common mode of activity such that activity becomes less odor-specific. This should be addressable by some simple analysis, and discussed appropriately in a few sentences.

As we understand this, there are two concerns about the background pattern.

First, there is the nature of the origin of increased correlations in OB under anesthesia. We have now included a new set of analyses that show that there is indeed an activity pattern – a population of weak but reliable and non-selective odor responses, i.e. a background pattern – which drives higher correlations under anesthesia. This occurs in both OB and PCx but is much more pronounced in OB. This was apparent in Figure 1l, but we have now explicitly demonstrated this effect in additional materials (Figure 1—figure supplement 5A-F), in which we subtracted the average odorevoked responses and find that residual responses are no longer correlated across odors.

Second, and most crucially, there is the question about whether this background pattern explains our main result: i.e. the preservation of PCx odor representations despite degraded OB representations. We stand by our original contention that OB odor responses degrade because some strong, odor-specific responses drop off under k/x anesthesia. We believe that we have demonstrated with multiple methods that this the degradation of OB odor representations is not dependent on the presence or absence of a background pattern in anesthetized OB:

- First, the ‘correlation separation’ shown in Figure 1m measures within-odor correlations that are above and beyond those expected from any across-odor correlation. Here, we directly subtracted off the across-odor correlation that is produced by background patterns in this analysis and show that separation is higher in PCx than OB.

- Second, any background pattern that is common across stimuli would not survive training of our SVM classifier, or any classifier. Cells that respond to all odors would have 0-weights in determining the classification boundary. The degraded classifier performance in OB under anesthesia (Figure 1N) demonstrates that it is difficult to find cells that respond consistently differentially to at least some subset of our odors.

- Third, in the dPCA analysis in Figure 2G and H), we explicitly removed overall statedependent patterns from our data. These are defined on the odor-evoked responses, so they effectively do remove any state-specific background patterns. Here again, we see PCx outperforming OB in maintaining similarity between awake and anesthetized responses.

- Finally, in the new material provided, we find that after explicitly subtracting state-specific background patterns before our correlation analyses, there remain weakened within-odor correlations in anesthetized OB in comparison with awake OB or awake or anesthetized PCx (Figure 1—figure supplement 5G,I), demonstrating again that there are less reliable odor representations in OB under anesthesia, while they remain more reliable in PCx.

In summary, two separate phenomena that occur under k/x anesthesia. First, there is an odor-dependent but not odor-specific component that increases within- and across-odor response correlations in both OB and PCx. On top of this, there is a pronounced degradation of OB responses due to a decrease in firing of some strong odor-specific responses that PCx is nevertheless able to accommodate.

Reviewer comments are provided in full below in case they are helpful.Reviewer #3):The authors have addressed some but not all comments.Comments:1) The most important issue is that the authors have not elucidated how anesthesia "degraded" odor representations in the OB. They observe that across-trial correlations go up, which tells us that a decrease in signal or an increase in noise cannot easily account for the effect of anesthesia. Their explanation that "degraded" decoding is due to reduced responses is inconsistent with the observation that all correlations increase (if activity is weakened and the signal to noise ratio decreases, correlations should converge to zero. If signal to noise ratio increases, which is possible when anesthesia reduces baseline activity, the difference between within and across odor correlations should increase, not decrease). An obvious possibility is that odor stimulation evokes a non-specific activity pattern under anesthesia that is independent of odor identity. This needs to be tested, as was put forward under main point 1 in the previous review. The authors have addressed this point incompletely because they only tested whether a background pattern present during spontaneous activity could increase correlations during odor responses (by subtracting mean spontaneous activity), but they did not test whether a non-specific "background" pattern is evoked by odor stimulation. There are many obvious ways to test this possibility (identifying a common mode in patterns by PCA or so and subtracting it out, or asking whether it is always the same cells that are responsible for different across-odor correlations, etc). As pointed out before, this issue is important because understanding the effects in the OB – which are obviously not a simple noise increase – is essential to understand the effects in PCx. Most predictions related to attractor networks consider inputs with some form of "random" noise, but anesthesia does not seem to induce "random" noise.

As stated above, this concern is two-pronged: first, why do response correlations increase, and second, do these increases in response correlations underlie/undermine our main conclusion that OB responses degrade under k/x anesthesia while PCx responses are comparatively robust?

Indeed, we do find that weak but reliable and non-specific odor responses emerge under k/x anesthesia, and these are especially pronounced in OB. These non-specific responses increase both within- and across-odor response correlations (Figure 1—figure supplement 5A-F).

However, although this background pattern increases correlations, it does not explain or trivialize the observation that PCx responses remain relatively robust under k/x anesthesia while OB responses degrade. Any ‘background pattern’ would be manifested in the average response across all odors. To test whether a non-specific ‘background’ pattern is the cause of the degraded representations in OB under anesthesia, we therefore computed and subtracted this average for both OB and PCx in awake and k/x trials, and then re-examined our correlation results using the residuals. This new analysis removes all non-odor-specific patterns of activity and clearly shows that although responses in OB degrade under k/x anesthesia while responses in PCx remain robust within- vs. across-odor (Figure 1—figure supplement 5G-I).

As an aside, there is no requirement in the definition of a fixed-point attractor that the attractor be perturbed by random noise. The attractor dynamics simply demand that specific points in state space are stable while others are not, such that, when the system is initiated to any unstable state – be it a random state or one caused by the superimposition of a stimulus-specific pattern with a background pattern – it will rapidly evolve toward a stable state and then remain there until perturbed again. Well-known examples of putative attractor-like dynamics do not assume ‘random’ noise. Rather, they often consist of morphing experiments, such as those suggested by reviewer 1, where an input pattern is subtly perturbed and recovers the original rather than an intermediate representation. Our study has much in common with Neunuebel et al., 2014, wherein DG and CA3 are monitored simultaneously and a cue manipulation is used to ‘degrade’ DG inputs to CA3. The population vector representations they observe at each degree position in their radial maze are analogous to the distinct odors in our experiments, and across-degree correlations can be quite high in DG under ‘degraded’ conditions, as are across-odor conditions in OB in our anesthetized condition. Yet CA3 population vectors maintain well-separated representations for each degree-bin despite degraded DG input. This is very like the sense in which we are interpreting these data as attractor-like, and we are cautious in our description because the system fails to perfectly recover this hypothetical stable state in our experiments. Nevertheless, the system moves in the direction of the known awake odor-specific state, consistent with a partial pattern recovery.

2) Along the same lines: Subtracting background rates did have a clear effect on accuracy (probably significant; please test) that is somewhat understated in the manuscript. It increases with the number of cells, as expected for a "background pattern". Is this effect enhanced during odor stimulation?

Subtracting either baseline firing or average odor-evoked firing patterns appeared to increase decoding accuracy slightly, however neither of these improvements was significant, as determined by confidence intervals between raw vs. subtracted performance across multiple permutations (Figure 1—figure supplement 7).

3) Along the same lines: the horizontal bands in Figure 2D, left panel, have not been explained. Please test whether they come from a background pattern (that may be observed only during odor stimulation, or enhanced by odor stimulation). If not, what else could it be?

The horizontal bands in Figure 2D occur when some awake trials have either more (light bands) or less (dark bands) correlated activity with all anesthetized trials. For the most part, these are due to some ‘early’ awake trials, which have had stronger responses (i.e. higher firing rates. Note that for these analyses we compared trials 6-13. Note also that PCx responses stabilized rapidly (i.e. within ~5 trials) whereas OB responses exhibited a slower and more modest trial-dependent adaptation (see Figure 6), and so many of the dark bands especially evident in OB confusion matrix can be explained by slightly higher overall firing rates during early OB trials. We have explicated this briefly in the Figure 2 legend.

4) The components in activity patterns responsible for increased across-odor correlations in piriform cortex under anesthesia have not been identified. These across-odor correlations are not predicted by attractor networks. They seem to argue against attractor networks. The authors should try to identify the sources for these correlations and address in the Discussion section whether or not they are consistent with an attractor network hypothesis.

Like OB, the increase in response correlations is due to a small subset of highly reliable, nonspecific odor responses. However, these are considerably less pronounced in PCx than OB, as evidenced by the greater separability of PCx vs. OB responses in awake vs. k/x trials.

5) The question whether the gradual stabilization of odor responses is due to plasticity of responses in the OB or PCx can be addressed in TeLC-PC mice: if the site of pasticity is PCx, stabilization of responses in the OB should be absent. Figure 6K seems to support this possibility but showing more data for TeLC-PC mice (as in Figure 6G) could clarify this question further.

We were able to analyze and observe stabilization across trials at the population level for individual PCx experiments because each PCx recording yielded a substantial number of simultaneously recorded neurons. Yields in OB recordings were often much smaller and would produce a very noisy and difficult to interpret outcome tracking the dynamics of stabilization on an individual experiment basis. These experiments would thus require an additional series of experiments in which we first inject TeLC into PCx and then recorded from multiple probes targeted to ipsi- and contralateral OB before and after k/x administration. While this is an interesting point, it is a little tangential, and we are presently months away from being able to even start these experiments. We therefore think that this question is beyond the scope of this study.

6) In Emx1-Cre mice, decoding seems to be even improved under k/x as compared to awake (compare Figure 4N to 4M), both in control and in TeLC expressing mice. Please explain.

Yes, the reviewer is correct, and we cannot really explain this observation. We have noticed previously (Bolding and Franks, 2018) that after TeLC expression, responses in the so-called contralateral control hemisphere are similar yet slightly different (i.e. ‘worse’) than true, unperturbed control PCx. We suspect this could be due to the modest contralateral projections, either directly from PCx or indirectly via AON or OB.

7) The conclusion that robust responses are actively amplified within PCx appears to strong because it is based only on a rather crude analysis of peak firing rates.

We disagree with the reviewer – peak firing rates (more accurately, spike counts within the first sniff) seem the most appropriate measure for whether or not a response is robust (see also Miura et al., 2012; Bolding and Franks, 2017).

8) It would be interesting to see correlation matrices such as Figure 1K that are generated only from SL or PYR cells.

Yes, however a detailed characterization of responses in SL vs. PYR cells is ongoing in the lab, will be described in another manuscript by Ms. Nagappan, and is not immediately relevant to the present study.